# Nonsmooth Implicit Differentiation for Machine Learning and Optimization

**Jérôme Bolte**
Toulouse School
of Economics
Univ. Toulouse
Toulouse, France

**Tam Le**
Toulouse School
of Economics
Univ. Toulouse
Toulouse, France

**Edouard Pauwels**
IRIT, CNRS
Univ. Toulouse
Toulouse, France

**Antonio Silveti-Falls**
Toulouse School
of Economics
Univ. Toulouse
Toulouse, France

## Abstract

In view of training increasingly complex learning architectures, we establish a non-smooth implicit function theorem with an operational calculus. Our result applies to most practical problems (i.e., definable problems) provided that a nonsmooth form of the classical invertibility condition is fulfilled. This approach allows for *formal subdifferentiation*: for instance, replacing derivatives by Clarke Jacobians in the usual differentiation formulas is fully justified for a wide class of non-smooth problems. Moreover this calculus is entirely compatible with algorithmic differentiation (e.g., backpropagation). We provide several applications such as training deep equilibrium networks, training neural nets with conic optimization layers, or hyperparameter-tuning for nonsmooth Lasso-type models. To show the sharpness of our assumptions, we present numerical experiments showcasing the extremely pathological gradient dynamics one can encounter when applying implicit algorithmic differentiation without any hypothesis.

## 1 Introduction

**Differentiable programming.**    The recent introduction of deep equilibrium networks [7], the increasing importance of bilevel programming (e.g., hyperparameter optimization) [48] and the ubiquity of differentiable programming (e.g., TensorFlow [1], PyTorch [47], JAX [16]) in modern optimization call for the development of a versatile theory of nonsmooth differentiation. Our focus is on nonsmooth implicit differentiation. There are currently two practices lying at the crossroads of mathematics and computer science: on the one hand the use of the standard smooth implicit function theorem "almost everywhere" [31, 30] and on the other hand the development of algorithmic differentiation tools [2, 3, 59]. The empirical use of the latter in the nonsmooth world has shown surprisingly efficient results [59], but the current theories cannot explain this success. We bridge this gap by providing nonsmooth implicit differentiation results and illustrating their impact on the training of neural networks and hyperparameter optimization.

**Backpropagation: a formal differentiation approach.**    Let us consider $z$ implicitly defined through $F(z(x)) = h(x)$ where $F$ and $h$ have full domain and adequate dimensions. How does autograd apply to evaluating the "derivative" of the implicitly defined function $z$? Regardless of differentiability or nonsmoothness, and provided that inversion is possible, one commonly uses (or dynamically approximates) this derivative by

$$(\mathrm{backprop}_F(z(x)))^{-1} \, \mathrm{backprop}_h x,$$

35th Conference on Neural Information Processing Systems (NeurIPS 2021).

where $\mathrm{backprop}$ outputs the result of formal backpropagation, see e.g., [50]. This identity[1] is used to provide efficient training despite the fact that the rules of classical nonsmooth calculus are transgressed [7, 59]. Note that spurious outputs may be created by this approach, but on a negligible set. Consider for example the simple implicit problem $x = f(z(x))$ where $f(z) := \tanh(z) + \mathrm{relu}(-z) + z - \mathrm{relu}(z)$, whose solution is $z(x) = \tanh x$. Yet applying the implicit differentiation framework of [7] using $\mathrm{JAX}$ library, as presented in [59], provides inconsistency of the derivative at the origin, see Figure 1. As mentioned above, despite these unpredictable outputs, propagating derivatives leads to an undeniable efficiency. But can we parallel these propagation ideas with a simple mathematical counterpart? Is there a rigorous theory backing up *formal (sub)differentiation* or *formal propagation*? The answer is positive and was initiated in [14, 15] through conservative Jacobians (see also [41, 24]).

**A mathematical model for propagating derivatives.** Conservative calculus models nonsmooth algorithmic differentiation faithfully and allows for a sharp study of training methods in Deep Learning [14, 15]. It involves a new class of derivatives, generalizing Clarke Jacobians [20]. A distinctive feature of conservative calculus is that it is preserved by Jacobian multiplication. Consider for example a feed forward network combining analytic or relu activations and max pooling. A conservative Jacobian for this network can be obtained by using Clarke Jacobians formally as classical Jacobians, regardless of qualification conditions, compared to other approaches, e.g. [37], for which stricter qualification conditions are imposed to ensure that an element of the Clarke Jacobian itself can be computed using algorithmic differentiation. For instance, Figure 1 depicts a selection in a conservative Jacobian. This approach is general enough to handle spurious points such as in Figure 1 while keeping the essence of the properties one expects from a derivative. It was proved in [14] that $\mathrm{backprop}$, applied to any reasonable program of a function, is a conservative Jacobian for this function; in contrast, $\mathrm{backprop}$ cannot be modelled by some subdifferential operator. For instance for the fixed point problem above, given conservative Jacobians $J_F$ and $J_h$ (e.g., Clarke Jacobians) for $F$ and $h$ one obtains a new conservative Jacobian $J_z$ implicitly defined through

$$J_F(z(x))J_z(x) = J_h(x).$$

This property exactly parallels the idea of "propagating derivatives" in practice. It gives a strong meaning to the formal use of Jacobians proposed in [7], and many empirical approaches [32, 2, 31, 30].

**Main contributions:**
— We establish a nonsmooth conservative implicit function theorem that comes with an *implicit calculus* which is the central focus of this paper. Our calculus amounts somehow to *formal subdifferentiation with Clarke Jacobians*. This approach cannot rely on classical tools like the inverse of a Clarke Jacobian or a composition of Clarke Jacobians, which are not in general Clarke Jacobians. Indeed, a surprising example (Example 1) shows that an "inverse function theorem with Clarke calculus" is not possible.

— We study a wide range of applications of our implicit differentiation theorem, covering deep equilibrium problems [7], conic optimization layers [2], and hyperparameter optimization for the Lasso [9]. Each case is detailed and its specificities are discussed.

— As a consequence, we obtain convergence guarantees for mini-batched stochastic algorithms with vanishing step size for training wide classes of Neural Nets, or for Lasso hyperparameter selection. The assumptions needed for our results are mild and fulfilled by most losses occurring in ML in the spirit of [15, 40]: elementary log-exp functions [15], semialgebraic functions [12], all being subclasses of definable functions [23, 55]. The use of such structural classes has become standard in nonsmooth optimization and is more and more common in ML (see, e.g., [18, 15, 40, 35]).

— As in the smooth implicit function theorem, the invertibility condition is not avoidable in general. We provide various examples for which the assumption is not satisfied; this results in severe failures for the corresponding gradient methods. In Figure 1, one sees how lack of invertibility on an otherwise ordinary problem may provide totally unpredictable behavior for smooth quadratic optimization.

**Related work on implicit differentiation:** The classical implicit function theorem has two parts: an existence and regularity part, and a calculus part, which is called implicit differentiation. Nons-

---

[1] The notation $\mathrm{backprop}_z$ instead of $\mathrm{backprop}(z)$ is indicative of the fact that $\mathrm{backprop}$ is an operator that does not act on functions themselves but rather on the program used to represent them, see [15].

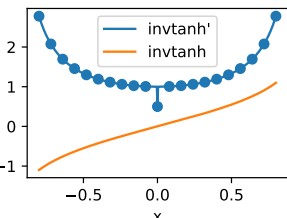 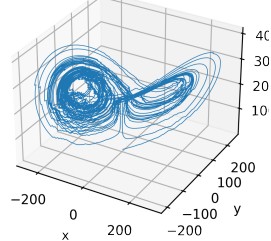

Figure 1: Left: Inconsistencies due to combination of implicit differentiation and algorithmic differentiation. Right: A gradient trajectory of an implicitly defined quadratic function.

mooth generalizations have mostly focused on the existence part. We focus on Lipschitz equations, for which existence of an implicit functional relation is due to [34], based on Clarke's inverse mapping theorem [19], all these elements being summarized in [20]. Various extensions of this result have been proposed, let us mention a result in an approximation context in [36], a semismooth extension in [52] and a tame extension in [29]; however, all these extensions lack a calculus amenable to implicit differentiation in practice. There have been extensions of the nonsmooth implicit function theorem with a calculus limited to directional derivatives and lexicographic derivatives (a sort of directional derivative introduced in [45]) in [49, Corollary 3.4] and [38, Theorem 2.3], respectively; the former requiring stronger assumptions than just the invertiblity condition. The limitation of the calculus of these extensions to directional derivatives is furthermore incompatible with the most common numerical libraries used for algorithmic differentiation in practice, as opposed to the framework we present.

**Definitions and Notations.** A function $F : \mathbb{R}^n \to \mathbb{R}^m$ is *locally Lipschitz* if, for each $x \in \mathbb{R}^n$, there exists a neighborhood $\mathcal{U}$ of $x$ such that $F$ is Lipschitz on $\mathcal{U}$. Given matrices $A \in \mathbb{R}^{n \times m}$ and $B \in \mathbb{R}^{n \times p}$, $[A\ B] \in \mathbb{R}^{n \times (m+p)}$ denotes their concatenation; $\mathrm{Id}_n$ denotes the $n \times n$ identity matrix. For $q \in \mathbb{R}^n$, $\mathrm{diag}\,(q) \in \mathbb{R}^{n \times n}$ denotes the diagonal matrix whose diagonal entries are given by the $q_i$; $\mathrm{sign}\,(q) \in \{-1, 0, 1\}^n$ denotes the componentwise sign function. The *convex hull* of $\mathcal{U}$ is denoted $\mathrm{conv}\,\mathcal{U}$. The projection onto a closed convex set $\mathcal{C} \in \mathbb{R}^n$ is given, for each $x \in \mathbb{R}^n$, by $P_{\mathcal{C}}\,(x) := \mathrm{argmin}\{\frac{1}{2}\,\|u - x\|^2 : u \in \mathcal{C}\}$. Given a convex proper lower semicontinuous function $f : \mathbb{R}^n \to \mathbb{R} \cup \{+\infty\}$, we define its proximal operator through $x \in \mathbb{R}^n$, $\mathrm{prox}_f\,(x) := \mathrm{argmin}\{f\,(u) + \frac{1}{2}\,\|u - x\|^2\ u \in \mathbb{R}^n\}$. Set-valued maps are denoted by $\rightrightarrows$, for example the subgradient $\partial f : \mathbb{R}^n \rightrightarrows \mathbb{R}^n$. Additional details and notations are provided in Appendix A.

## 2 Implicit Differentiation with Conservative Jacobians

**Definitions and conservativity.** Conservative Jacobians are generalized forms of Jacobians well suited for automatic differentiation, introduced in [14]. Given a locally Lipschitz continuous function $F : \mathbb{R}^n \to \mathbb{R}^m$, we say that $J_F : \mathbb{R}^n \rightrightarrows \mathbb{R}^{n \times m}$ is a *conservative mapping* or a *conservative Jacobian* for $F$ if $J_F$ has a closed graph, is locally bounded, and is nonempty with

$$\frac{\mathrm{d}}{\mathrm{d}t} F(\gamma(t)) \in J_F(\gamma(t))\dot{\gamma}(t) \text{ a.e.} \tag{1}$$

whenever $\gamma$ is an absolutely continuous curve in $\mathbb{R}^n$. When $m = 1$, the corresponding vectors are called *conservative gradient fields*. Note that when $J_F$ is conservative, so is its pointwise convexified extension $\mathrm{conv}\,J_F$.

A locally Lipschitz function is called *path differentiable* if it has a conservative Jacobian. Recall that the *Clarke Jacobian* is defined as

$$\mathrm{Jac}^c F(x) = \mathrm{conv} \left\{ \lim_{k \to +\infty} \mathrm{Jac}\, F(x_k) : x_k \in \mathrm{diff}_F, x_k \xrightarrow[k \to +\infty]{} x \right\}$$

where $\mathrm{diff}_F$ is the full measure set of points where $F$ is differentiable and $\mathrm{Jac}\, F$ is the standard Jacobian of $F$. Path differentiability is equivalent to having a chain rule as in (1) for the Clarke subdifferential, see [14, 25].

**Examples of path differentiable functions and conservative Jacobians.** (a) Convex functions and concave functions are path differentiable, see [14]. This implies that their subdifferential in the sense of convex analysis is a conservative field.

(b) The vast class of definable functions are path differentiable [25, 14]. As a result, the Clarke Jacobian of a Lipschitz definable mapping is a conservative Jacobian. Definable functions (see [5, 25, 18, 14] for an optimization context and [23] for a foundational work) encompass semialgebraic functions [12], elementary log-exp selection [14], PAP [40] (restricted to analytic functions with full domain), and many others, see [55] and references therein. This includes networks with common nonlinearities: for example analytic with full domain (e.g., square, exponential, logistic loss, hyperbolic tangent, sigmoid), relu, max pooling, sort, (see Appendix A.2 for more detail).

(c) The backpropagation can be seen as an oracle (in the optimization sense) for a conservative Jacobian. Let $P_F$ be a numerical program for a function $F$, aggregating elementary functions, for instance, relu, max pooling, affine mappings, polynomials (in general, any definable function). Then the back-propagation algorithm applied to $P_F$, which we denote (abusively) by $\mathrm{backprop}\, P_F := \mathrm{backprop}_F$, outputs an element of a conservative Jacobian [14, Theorem 8] which depends on $P_F$ and can be constructed by a closure procedure [15, definition 5]. As described in [15], due to spurious behaviors, $\mathrm{backprop}_F$ is not in general an element of the Clarke Jacobian of $F$.

**The structure of conservative Jacobians.** As established in [41] in a semialgebraic context, the discrepancy between conservative gradients and Clarke subdifferentials is somehow negligible. Let us provide a version of that result matching our concerns. We call conservative mappings of the null function *residual* or *residual conservative*. Such a mapping $R$ has the property that $R(x + tv)v = 0$ for almost all $t$ in $\mathbb{R}$ and all $x, v$ in $\mathbb{R}^n \times \mathbb{R}^n$. The following theorem and proposition (partially) extend results from [14] and [41], their proof is given in Appendix B.

**Theorem 1 (The Clarke Jacobian is a minimal conservative Jacobian)** *Given a nonempty open subset $\mathcal{U}$ of $\mathbb{R}^n$ and $F : \mathcal{U} \subset \mathbb{R}^n \to \mathbb{R}^m$ locally Lipschitz, let $J_F$ be a convex-valued conservative Jacobian for $F$. Then for almost all $x \in \mathcal{U}$, $J_F(x) = \{\mathrm{Jac}\, F\}$ and, for all $x \in \mathcal{U}$, $\mathrm{Jac}^{\,c}F(x) \subset J_F(x)$.*

**Proposition 1 (Decomposition of conservative fields)** *Let $J_F$ be a conservative Jacobian for $F$, then there is a residual $R$ such that*

$$J_F \subset \mathrm{Jac}^{\,c}F + R.$$

Note that the above may not hold with equality. Consider $F(x) = |x|$ and $J_F(0) = [-1, 1] \cup [2, 3]$, $J_F(x) = \mathrm{sign}\,(x)$ otherwise. One cannot write $J_F = \mathrm{Jac}^{\,c}F + R$ with a residual operator $R$.

**Formal subdifferentiation in a nonsmooth setting.** Propagating derivatives within a nonsmooth function finds its justification in the following:

**Proposition 2 (Stability by composition, [14])** *Let $F : \mathbb{R}^n \to \mathbb{R}^m$ and $G : \mathbb{R}^m \to \mathbb{R}^l$ be two locally path differentiable functions having respective conservative Jacobians $J_F$ and $J_G$. Then $F \circ G$ is path differentiable and the point-to-set matrix-valued $x \rightrightarrows J_F(G(x))J_G(x)$ is conservative.*

**A conservative Implicit Function Theorem.** There is already a long tradition of nonsmooth implicit function theorems, e.g., [20, 6, 49, 26]. What makes the following theorem useful is that it comes with a qualification-free calculus. The proofs are given in Appendix B.

**Theorem 2 (Implicit differentiation)** *Let $F : \mathbb{R}^n \times \mathbb{R}^m \to \mathbb{R}^m$ be path differentiable on $\mathcal{U} \times \mathcal{V} \subset \mathbb{R}^n \times \mathbb{R}^m$ an open set and $G : \mathcal{U} \to \mathcal{V}$ a locally Lipschitz function such that, for each $x \in \mathcal{U}$,*

$$F(x, G(x)) = 0. \tag{2}$$

*Furthermore, assume that for each $x \in \mathcal{U}$, for each $[A\ B] \in J_F(x, G(x))$, the matrix $B$ is invertible where $J_F$ is a conservative Jacobian for $F$. Then, $G : \mathcal{U} \to \mathcal{V}$ is path differentiable with conservative Jacobian given, for each $x \in \mathcal{U}$, by*

$$J_G \colon x \rightrightarrows \left\{ -B^{-1}A : [A\ B] \in J_F(x, G(x)) \right\}.$$

**Corollary 1 (Path differentiable implicit function theorem)** *Let $F : \mathbb{R}^n \times \mathbb{R}^m \to \mathbb{R}^m$ be path differentiable with conservative Jacobian $J_F$. Let $(\hat{x}, \hat{y}) \in \mathbb{R}^n \times \mathbb{R}^m$ be such that $F(\hat{x}, \hat{y}) = 0$. Assume that $J_F(\hat{x}, \hat{y})$ is convex and that, for each $[A \; B] \in J_F(\hat{x}, \hat{y})$, the matrix $B$ is invertible. Then, there exists an open neighborhood $\mathcal{U} \times \mathcal{V} \subset \mathbb{R}^n \times \mathbb{R}^m$ of $(\hat{x}, \hat{y})$ and a path differentiable function $G : \mathcal{U} \to \mathcal{V}$ such that the conclusion of Theorem 2 holds.*

**Corollary 2 (Path differentiable inverse function theorem)** *Let $\mathcal{U}$ and $\mathcal{V}$ be open neighborhoods of $0$ in $\mathbb{R}^n$ and $\Phi : \mathcal{U} \to \mathcal{V}$ path differentiable with $\Phi(0) = 0$. Assume that $\Phi$ has a conservative Jacobian $J_\Phi$ such that $J_\Phi(0)$ contains only invertible matrices. Then, locally, $\Phi$ has a path differentiable inverse $\Psi$ with a conservative Jacobian given by*

$$J_\Psi(y) = \left\{ A^{-1} : A \in J_\Phi(\Psi(y)) \right\}.$$

**Remark 1** (a) **(On the necessity of conservativity)** Example 1 in Appendix B shows that one cannot hope for the formulas in Corollaries 1 & 2 to provide Clarke Jacobians in general, even if the input(s) are Clarke Jacobians themselves. This example is bivariate and piece-wise linear, illustrating that even in this simple situation, implicit differentiation fails and produces artifacts when formaly applied to Clarke Jacobians.
(b) **(Lipschitz definable implicit and inverse function theorems)** See Theorem 4 and 5 in the appendix

## 3 Nonsmooth implicit differentiation in Machine Learning

Detailed proof arguments for all considered models are given in Appendix C.

**Monotone deep equilibrium networks.** Deep Equilibrium Networks (DEQs) [7] are specific neural network architectures including layers whose input-output relation is implicitly defined through a fixed point equation of the form

$$z = f(z, x) \tag{3}$$

where $x \in \mathbb{R}^p$ is a given input and $z \in \mathbb{R}^m$ is the corresponding output. We may consider that the variable $x$ represents both the input layer and layer parameters. Assuming that, for each $x \in \mathbb{R}^p$, there is a unique $z \in \mathbb{R}^m$ satisfying the relation (3), this defines an input-output relation $z : \mathbb{R}^p \to \mathbb{R}^m$. Furthermore, if $f$ is path differentiable with convex-valued conservative Jacobian $J_f : \mathbb{R}^m \times \mathbb{R}^p \rightrightarrows \mathbb{R}^{m \times (m+p)}$ whose projection on the first $m$ columns are all invertible, then the function $z$ itself admits a conservative Jacobian which can be computed from Theorem 2.

We now focus on monotone operator implicit layers [58] for which assumptions are easily stated. Our method applies to other similar architectures, e.g., DEQs [7] or implicit graph neural networks [32]. Let $\sigma : \mathbb{R}^m \to \mathbb{R}^m$ be the proximal operator of a convex function and assume $\sigma$ is path differentiable with conservative Jacobian $J_\sigma : \mathbb{R}^m \rightrightarrows \mathbb{R}^{m \times m}$, assumed to be convex-valued. This encompasses the majority of activation functions used in practice [21]. Let $W \in \mathbb{R}^{m \times m}$ be a matrix such that $W + W^T \succeq 2\theta I$ with $\theta > 0$. Under these assumptions the implicit equation

$$z = \sigma(Wz + b) \tag{4}$$

has a unique output $z(W, b)$ [58, Theorem 2]. The transformation $(W, b) \mapsto z(W, b)$ is a *monotone implicit layer*.

The set-valued mapping obtained from Theorem 2 provides a conservative Jacobian for $(W, z) \mapsto z(W, z)$. A similar expression was described in [58, Theorem 2], without using conservativity and using the Clarke Jacobian formally as a classical Jacobian. The proposition below provides a full justification of this heuristic and ensures convergence of algorithmic differentiation based training.

**Proposition 3 (Path differentiation through monotone layers)** *Assume that $J_\sigma$ is convex-valued and that, for all $J \in J_\sigma(Wz(W, b) + b)$, the matrix $(\mathrm{Id}_m - JW)$ is invertible. Consider a loss-like function $\ell : \mathbb{R}^m \to \mathbb{R}$ with conservative gradient $D_\ell : \mathbb{R}^m \rightrightarrows \mathbb{R}^m$, then $g : (W, z) \mapsto \ell(z(W, b))$ is path differentiable and has a conservative gradient $D_g$ defined through*

$$D_g : (W, b) \rightrightarrows \left\{ J^T (\mathrm{Id}_m - JW)^{-T} v z^T, \, J^T (\mathrm{Id}_m - JW)^{-T} v \, : \, J \in J_\sigma(Wz + b), \, v \in D_\ell(z) \right\}.$$

**Remark 2** Convexity and invertibility assumptions are satisfied when $J_\sigma$ is the Clarke Jacobian [58].

**Optimization layers: the conic program case.** Optimization layers in deep learning may take many forms; we consider here those based on conic programming [17, 3, 2, 4]. We follow [3], simplifying the analysis by ignoring infeasability certificates, which correspond to the absence of a primal-dual solution [17], in line with the implementation described in [2, Appendix B]. Consider a conic problem (P) and its dual (D):

$$
\text{(P)} \quad \begin{aligned} &\inf & & c^T x \\ &\text{subject to} & & Ax + s = b \\ & & & s \in \mathcal{K} \end{aligned} \qquad\qquad \text{(D)} \quad \begin{aligned} &\inf & & b^T y \\ &\text{subject to} & & A^T y + c = 0 \\ & & & y \in \mathcal{K}^*, \end{aligned} \qquad (5)
$$

with primal variable $x \in \mathbb{R}^n$, dual variable $y \in \mathbb{R}^m$, and primal slack variable $s \in \mathbb{R}^m$. The set $\mathcal{K} \subset \mathbb{R}^m$ is a nonempty closed convex cone and $\mathcal{K}^* \subset \mathbb{R}^m$ is its dual cone. The problem parameters are the matrix $A \in \mathbb{R}^{m \times n}$ and the vectors $b \in \mathbb{R}^m$ and $c \in \mathbb{R}^n$; the cone $\mathcal{K}$ is fixed. Under the assumption that there is a unique primal-dual solution $(x, y, s)$, we study the path differentiability of the solution mapping as a function of its parameters:

$$
(A, b, c) \mapsto \text{sol}(A, b, c) = (x, y, s).
$$

For this, let us interpret the solution mapping as a composition mapping involving equation-like implicit formulations. Set $N = n + m$, given $A, b, c \in \mathbb{R}^{m \times n} \times \mathbb{R}^m \times \mathbb{R}^n$, define

$$
Q(A, b, c) = \begin{bmatrix} 0 & A^T \\ -A & 0 \end{bmatrix} \in \mathbb{R}^{N \times N} \qquad V(b, c) = \begin{bmatrix} c \\ b \end{bmatrix} \in \mathbb{R}^N.
$$

Consider a vector $z = (u, v) \in \mathbb{R}^n \times \mathbb{R}^m$, denote by $\Pi$ the projection onto $\mathbb{R}^n \times \mathcal{K}^*$ and define the *residual map* $\mathcal{N} : \mathbb{R}^N \times \mathbb{R}^{m \times n} \times \mathbb{R}^m \times \mathbb{R}^n \to \mathbb{R}^N$ as

$$
\mathcal{N}(z, A, b, c) = (Q(A, b, c) - \text{Id}_N)\Pi z + V(b, c) + z.
$$

The mapping $\mathcal{N}$ is a synthetic form of optimality measure for (P) and (D), capturing KKT conditions. To simplify the presentation, we ignore the extreme cases of infeasibility and unboundedness which correspond to an absence of solution in [17].

Define the function $\phi : \mathbb{R}^N \to \mathbb{R}^n \times \mathbb{R}^m \times \mathbb{R}^m$ through $\phi(u, v) := (u, P_{\mathcal{K}^*}(v), P_{\mathcal{K}^*}(v) - v)$. As shown in Appendix C.2, $\phi(u, v)$ provides a primal-dual KKT solution of problems (P) and (D) if and only if $\mathcal{N}(z, A, b, c) = 0$. When we assume that, for fixed $A$, $b$, and $c$, there is a unique $z \in \mathbb{R}^N$ such that $\mathcal{N}(z, A, b, c) = 0$, we have an implicitly defined a function $z = \nu(A, b, c)$, such that

$$
\text{sol}(A, b, c) = [\phi \circ \nu](A, b, c). \qquad (6)
$$

The following result extends the discussion in [17, 3], limited to situations where $\Pi$ is differentiable at the proposed solution $z$, to a fully nonsmooth setting; its proof is postponed to Appendix C.2.

**Proposition 4 (Path differentiation through cone programming layers)** *Assume that $P_{\mathcal{K}^*}$, $\mathcal{N}$ are path differentiable, denote respectively by $J_{P_{\mathcal{K}^*}}$, $J_{\mathcal{N}}$ corresponding convex-valued conservative Jacobians. Assume that, for all $A, b, c \in \mathbb{R}^{m \times n} \times \mathbb{R}^m \times \mathbb{R}^n$, $z = \nu(A, b, c) \in \mathbb{R}^n \times \mathbb{R}^m$ is the unique solution to $\mathcal{N}(z, A, b, c) = 0$ and that all matrices formed from the $N$ first columns of $J_{\mathcal{N}}(z, A, b, c)$ are invertible. Then, $\phi$, $\nu$, and* sol *are path differentiable functions with conservative Jacobians:*

$$
J_\nu(A, b, c) := \left\{ -U^{-1}V : [U\ V] \in J_{\mathcal{N}}(\nu(A, b, c), A, b, c) \right\},
$$

$$
J_\phi(z) := \begin{bmatrix} \text{Id}_n & 0 \\ 0 & J_{P_{\mathcal{K}^*}}(v) \\ 0 & (J_{P_{\mathcal{K}^*}}(v) - \text{Id}_m) \end{bmatrix},
$$

$$
J_{\text{sol}}(A, b, c) := J_\phi(\nu(A, b, c)) J_\nu(A, b, c).
$$

In practice, the path differentiability of conic projections is pervasive since they are generally semialgebraic (orthant, second-order cone, PSD cone). See [33, 43, 39, 43] for the computations of the corresponding Clarke Jacobians (which are conservative). Note that a conservative Jacobian for $\mathcal{N}$ may be obtained from $J_{P_{\mathcal{K}^*}}$ using Proposition 2.

**Hyperparameter selection for Lasso type problems.** Implicit differentiation can be used to tune hyperparameters via first-order methods optimizing some measure of task performance, see [10] and references therein. In a nonsmooth context, we recall the formulation in [9] of the general hyperparameter optimization problem as a bi-level optimization problem:

$$\min_{\lambda \in \mathbb{R}^m} C(\hat{\beta}(\lambda)) \quad \text{such that} \quad \hat{\beta}(\lambda) \in \operatorname*{argmin}_{\beta \in \mathbb{R}^p} \psi(\beta, \lambda)$$

where $C : \mathbb{R}^p \to \mathbb{R}$ is continuously differentiable (e.g., test loss) and $\psi \colon \mathbb{R}^p \times \mathbb{R}^m \to \mathbb{R}$ is a possibly nonsmooth training loss, convex in $\beta$, with hyperparameter $\lambda \in \mathbb{R}^m$. We seek a subgradient type method for this problem with convergence guaranties; our nonsmooth implicit differentiation results can be used for this purpose. We demonstrate this approach on the Lasso problem [53]

$$\hat{\beta}(\lambda) \in \operatorname*{argmin} \left\{ \frac{1}{2} \|y - X\beta\|_2^2 + e^\lambda \|\beta\|_1 : \beta \in \mathbb{R}^p \right\} \tag{7}$$

where $y \in \mathbb{R}^n$ is the vector of observations, $X = [X_1, \ldots, X_p] \in \mathbb{R}^{n \times p}$ is the design matrix with columns $X_j \in \mathbb{R}^n$, $j \in \{1, \ldots p\}$, and $\lambda \in \mathbb{R}$ is the hyperparameter. Define $F : \mathbb{R} \times \mathbb{R}^p \to \mathbb{R}^p$ to be

$$F(\lambda, \beta) := \beta - \operatorname{prox}_{e^\lambda \|\cdot\|_1} \left( \beta - X^T (X\beta - y) \right)$$

and recall that, for each $i \in \{1, \ldots, p\}$, $[\operatorname{prox}_{e^\lambda \|\cdot\|_1}(\beta)]_i = \operatorname{sign}(\beta_i) \max\{|\beta_i| - e^\lambda, 0\}$. The function $F(\lambda, \beta)$ is thus nonsmooth but locally Lipschitz on $\mathbb{R} \times \mathbb{R}^p$. An optimal $\hat{\beta}(\lambda)$ for (7) must satisfy $F(\lambda, \hat{\beta}(\lambda)) = 0$ [22, Prop. 3.1]. For a given solution $\hat{\beta}(\lambda)$, we introduce the equicorrelation set by $\mathcal{E} := \{j \in \{1, \ldots, p\} : |X_j^T(y - X\hat{\beta}(\lambda))| = e^\lambda\}$ which contains the support set $\operatorname{supp} \hat{\beta} := \{i \in \{1, \ldots, p\} : \hat{\beta}_i \neq 0\}$. In fact, $\mathcal{E}$ does not depend on the choice of the solution $\hat{\beta}$, see [54, Lemma 1]. The proof of the following result is given in Appendix C.3.

**Proposition 5 (Conservative Jacobian for the solution mapping)** *For all $\lambda \in \mathbb{R}$, assume $X_\mathcal{E}^T X_\mathcal{E}$ is invertible where $X_\mathcal{E}$ is the submatrix of $X$ formed by taking the columns indexed by $\mathcal{E}$. Then $\hat{\beta}(\lambda)$ is single-valued, path differentiable with conservative Jacobian, $J_{\hat{\beta}}(\lambda)$, given for all $\lambda$ as*

$$\left\{ \left[ -e^\lambda \left( \operatorname{Id}_p - \operatorname{diag}(q)\left(\operatorname{Id}_p - X^T X\right) \right)^{-1} \operatorname{diag}(q) \operatorname{sign}\left(\hat{\beta} - X^T\left(X\hat{\beta} - y\right)\right) \right] : q \in \mathcal{M}(\lambda) \right\}$$

*where $\mathcal{M}(\lambda) \subset \mathbb{R}^p$ is the set of vectors $q$ such that $q_i = 1$ if $i \in \operatorname{supp} \hat{\beta}$, $q_i = 0$ if $i \notin \mathcal{E}$ and $q_i \in [0, 1]$ if $i \in \mathcal{E} \setminus \operatorname{supp} \hat{\beta}$.*

Taking, in Proposition 5, $q_i = 1$ for all $i \in \mathcal{E}$ corresponds to the directional derivative given by LARS algorithm [28], see also [42]. Alternatively, taking $q_i = 0$ for $i \notin \operatorname{supp} \hat{\beta}$ gives the weak derivative described by [9]. Both are particular selections in $J_{\hat{\beta}}$, which is the underlying conservative field.

## 4 Optimizing implicit problems with gradient descent

We establish the convergence of gradient descent algorithms for compositional learning problems involving implicitly defined functions. The result follows from the previous section and the general convergence results of [15].

**The minimization problem.** The applications considered in the previous section all yield minimization problems of the type

$$\min_{w \in \mathbb{R}^p} \ell(w) := \frac{1}{N} \sum_{i=1}^N \ell_i(w) \text{ with } \ell_i = g_{i,L} \circ g_{i,L-1} \circ \ldots \circ g_{i,1} \tag{8}$$

where, for each $i \in \{1, \ldots, N\}$, $\ell_i : \mathbb{R}^p \to \mathbb{R}$ is a composition of functions having appropriate input and output dimensions. The indices $i$ correspond in practice to learning samples while the loss $\ell$ embodies an empirical expectation, as for instance in deep learning. We will enforce the following structural condition.

**Assumption 1** *For $i \in \{1, \ldots, N\}$ and $j \in \{1, \ldots, L\}$, the function $g_{i,j}$ is locally Lipschitz with conservative Jacobian $J_{i,j}$ and one of the following holds*

- $g_{i,j}$ *and $J_{i,j}$ are semialgebraic (or, more generally, definable).*

- $g_{i,j}$ *is defined as $G$ in Theorem 2, with $F$ and $J_F$ semialgebraic (or, more generally, definable).*

Actually, in Assumption 1 the second point implies the first point; we list both for clarity. More details on semialgebraicity and definability are given in Appendix A.2. Let us stress that virtually all elements entering the definition of neural networks are semialgebraic or, more generally, definable, see for example [15] for a constructive model. In particular, beyond classical networks with usual nonlinearities (e.g., relu, sigmoid, max pooling ... ), this setting encompasses (through Corollary 1):

**(a)** Deep equilibrium networks: each $g_{i,j}$ may correspond to usual explicit layers or an implicit layer involving a fixed point mapping and a learning sample $i$ as in (4) or (3).

**(b)** Training with optimization layers: similarly, the inner maps $g_{i,j}$ may also be solution mapping to convex conic programs and related to the sol function (6) of conic problems.

**(c)** One may assume that $N = 1, L = 2$ and retrieve the hyperparameter tuning for Lasso in its implicit formulation.

**SGD with backpropagation.** Algorithmic differentiation (AD) is an automated application of the chain rule of differential calculus. When applied to $\ell_i$, it amounts to computing one element of the product $J_i := \prod_{j=1}^{L} J_{i,j}$ by choosing one element in each $J_{i,j}$ with appropriate inputs given by intermediate results kept in memory during a forward computation of the composition.

In this context AD stochastic gradient descent requires an initial $w_0 \in \mathbb{R}^p$ and a sequence of *i.i.d.* random indices uniform in $\{1, \ldots, N\}$, $(I_k)_{k \in \mathbb{N}}$. It gives:

$$w_{k+1} = w_k - s\alpha_k v_k \tag{9}$$

$$v_k \in J_{I_k}(w_k), \qquad \text{(given by backprop)}, \tag{10}$$

where $(\alpha_k)_{k \in \mathbb{N}}$ is a sequence of positive step sizes and $s \in (s_{\min}, s_{\max})$ is a scaling factor where $s_{\max} > s_{\min} > 0$. A simpler choice could be $v_k \in \partial^c \ell_{I_k}(w_k)$, however, the chain rule used within algorithmic differentiation routines does not produce subgradients (see, e.g., Figure 1). In contrast, conservative Jacobians are faithful models of AD outputs. The asymptotic behavior of the above algorithm depends on the variational properties of the conservative Jacobian $J := \frac{1}{N} \sum_{i=1}^{N} J_i$.

**Theorem 3 (Convergence result)** *Consider minimizing $\ell$ given in (8) using algorithm (10) under Assumption 1. Assume furthermore the following*

- ***Step size:*** $\sum_{k=1}^{+\infty} \alpha_k = +\infty$ *and $\alpha_k = o(1/\log(k))$.*

- ***Boundedness:*** *there exists $M > 0$, and $K \subset \mathbb{R}^p$ open and bounded, such that, for all $s \in (s_{\min}, s_{\max})$ and $w_0 \in \mathrm{cl}\, K$, $\|w_k\| \leq M$ almost surely.*

*For almost all $w_0 \in K$ and $s \in (s_{\min}, s_{\max})$, the objective value $\ell(w_k)$ converges and all accumulation points $\bar{w}$ of $w_k$ are Clarke-critical in the sense that $0 \in \partial^c \ell(\bar{w})$.*

This result shows that AD SGD may be applied successfully to all problems described in Section 3, combining algorithmic differentiation with implicit differentiation. Its proof may be adapted directly from [14, 11]; details are given in Appendix D.

## 5 Numerical experiments

Using implicit differentiation when the invertibility condition in Theorem 2 does not hold can result in absurd training dynamics.

**A cyclic gradient dynamics via fixed-point/optimization layer.** Consider the bilevel problem:

$$\min_{x,y,s} \quad \ell(x, y, s) := (x - s_1)^2 + 4(y - s_2)^2 \tag{11}$$

$$\text{s.t.} \quad s \in s(x, y) := \arg\max \left\{ (a + b)(-3x + y + 2) : a \in [0, 3], b \in [0, 5] \right\}.$$

Problem (11) has an equivalent fixed-point formulation using projected gradient descent on the inner problem (Appendix E.1.1). Backpropagation applied to (11) associates to $(x, y)$ the following:

$$\nabla_{(x,y)}\ell(x, y, s(x)) + \tilde{J}_s(x, y)^T \nabla_s \ell(x, y, s(x)) \tag{12}$$

where $\tilde{J}_s$ is piecewise derivative. We implement gradient descent for (11), evaluating (12) either using `cvxpylayers` [2] or the `JAX` tutorial [59] for fixed-point layers. In both cases, the invertibility condition in Theorem 2 fails when $-3x + y + 2 = 0$, resulting in discontinuity of $s$, affecting the dynamics globally: the gradient trajectory converges to a limit cycle of non critical points (Figure 2a); see Appendix E.1 for details.

*Persistence under small perturbations:* For different initial points the gradient flow converges to the same limit cycle (Figure 2a). The cycle persists even if we perturb the coefficients in the problem (11) ( see Appendix E.1.2 for more details) .

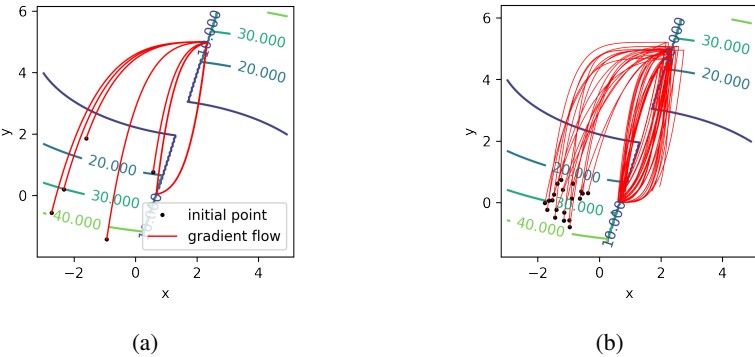

|     |     |
| :-: | :-: |
| (a) | (b) |

Figure 2: (a) Gradient flow for several initializations. (b) Gradient flow for 20 perturbed experiments with $\sigma^2 = 0.4$.

**A Lorenz-like dynamics via implicit differentiation.**   The Lorenz Ordinary Differential Equation (ODE) writes:

$$\dot{x} = \sigma(y - x), \qquad \dot{y} = x(\rho - z) - y, \quad \text{and} \quad \dot{z} = xy - \beta z. \tag{13}$$

It is well-known that taking $(\sigma, \rho, \beta) = (10, 28, 8/3)$, and $(x(0), y(0), z(0)) = (0, 1, 1.05)$ gives a chaotic trajectory, displayed in Figure 3a. Denoting $F : (x, y, z) \mapsto (\sigma(y-x), x(\rho-z)-y, xy-\beta z)$ the vector field of the Lorenz system (13), consider the optimization problem:

$$\max_{u \in \mathbb{R}^3} \quad u^T z \qquad \text{s.t.} \qquad z \in \arg\min_{s \in \mathbb{R}^3} \|s - F(u)\|^4 \tag{14}$$

which is obviously equivalent to

$$\max_{u \in \mathbb{R}^3} \quad u^T F(u). \tag{15}$$

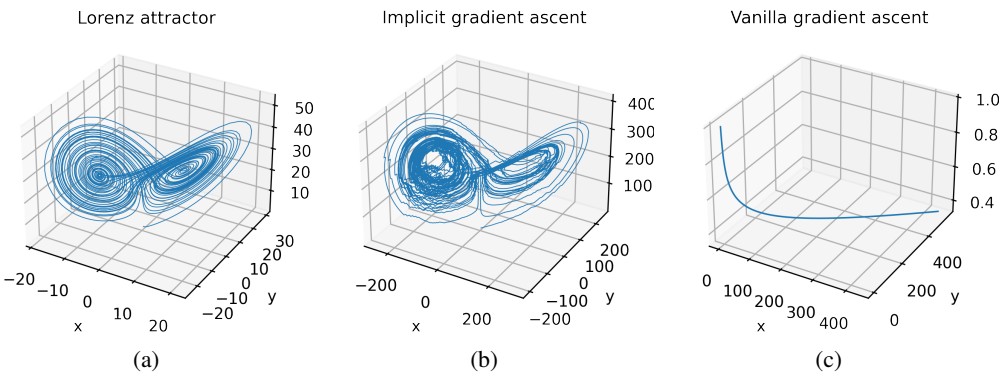

|     |     |     |
| :-: | :-: | :-: |
| (a) | (b) | (c) |

Figure 3: Implicit gradient ascent (b) outputs a pathological curve with some qualitative aspects of the Lorenz dynamics (a) and really different from a classical gradient (c).

The function $g : u \mapsto u^T F(u)$ is a nondegenerate quadratic function whose expression can be found in Appendix E.2.1. The function $g$ has for unique critical point $(0, 0, 0)$ which is a strict saddle-point. We perform gradient ascent with implicit differentiation using `cvxpylayers` on (14), and the classical gradient ascent on the equivalent problem (15). The path obtained by implicit differentiation (Figure 3b) resembles the Lorenz attractor (Figure 3a), in stark contrast to the conventional method (Figure 3c). The chaotic dynamics are a consequence of the lack of invertibility, due to the power $4$ in (14), and various numerical approximations related to optimization and implicit differentiation.

## 6    Conclusion and future work

This article provides a rigorous framework and calculus rules for nonsmooth implicit differentiation using the theory of conservative Jacobians. In particular, it describes precise conditions under which implicit differentiation can be used, in a way that is compatible with backpropagation and first-order algorithms.

We show the applicability of our results on practical machine learning problems including training of neural networks involving layers with implicitly defined outputs (deep equilibrium nets, networks with optimization layers) and nonsmooth hyperparameter optimization (Lasso-type models).

Finally, we demonstrate the necessity of a rigorous theory of nonsmooth implicit differentiation through multiple numerical experiments. These illustrate the range of extremely pathological gradient dynamics that can occur when algorithmic differentiation is combined with nonsmooth implicit differentiation outside the scope of our theorem, i.e., without satisfying the invertibility condition we specify.

## Acknowledgments and Disclosure of Funding

The authors acknowledge the support of AI Interdisciplinary Institute ANITI funding, through the French "Investing for the Future – PIA3" program under the Grant agreement ANR-19-PI3A-0004, Air Force Office of Scientific Research, Air Force Material Command, USAF, under grant numbers FA9550-19-1-7026, FA9550-18-1-0226, and ANR MaSDOL 19-CE23-0017-01. J. Bolte also acknowledges the support of ANR Chess, grant ANR-17-EURE-0010 and TSE-P.

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
