This is the appendix for "Nonsmooth Implicit Differentiation for Machine Learning and Optimization".

## Appendices

## A Lexicon

### A.1 Conservative fields

We first collect the necessary definitions to define a conservative set-valued field, introduced in [14], and by extension conservative Jacobians. Recall from multivariable calculus that the *Jacobian* of a differentiable function $f : \mathbb{R}^n \to \mathbb{R}^m$ is given by

$$
\operatorname{Jac} f := \begin{bmatrix} \frac{\partial f_1}{\partial x_1} & \cdots & \frac{\partial f_1}{\partial x_n} \\ \vdots & \ddots & \vdots \\ \frac{\partial f_m}{\partial x_1} & \cdots & \frac{\partial f_m}{\partial x_n} \end{bmatrix}.
$$

**Definition 1 (Absolutely continuous curve)** *A continuous function $\gamma : \mathbb{R} \to \mathbb{R}^n$ is an absolutely continuous curve if it has a derivative $\dot{\gamma}(t)$, for almost all $t \in \mathbb{R}$, which furthermore satisfies*

$$
\gamma(t) - \gamma(0) = \int_0^t \dot{\gamma}(\tau)d\tau
$$

*for all $t \in \mathbb{R}$.*

The *graph* of a set-valued mapping $D : \mathbb{R}^n \rightrightarrows \mathbb{R}^m$ is the set $\operatorname{graph} D := \{(x, z) : x \in \mathbb{R}^n, z \in D(x)\}$.

**Definition 2 (Closed graph)** *A set-valued mapping $D : \mathbb{R}^n \rightrightarrows \mathbb{R}^m$ has closed graph or is graph closed if $\operatorname{graph} D$ is a closed subset of $\mathbb{R}^{n+m}$ or, equivalently, if, for any convergent sequences $(x_k)_{k \in \mathbb{N}}$ and $(z_k)_{k \in \mathbb{N}}$ with $z_k \in D(x_k)$ for all $k \in \mathbb{N}$, it holds*

$$
\lim_{k \to \infty} z_k \in D\left(\lim_{k \to \infty} x_k\right).
$$

**Definition 3 (Locally bounded)** *A set-valued mapping $D : \mathbb{R}^n \rightrightarrows \mathbb{R}^m$ is locally bounded if for all $x \in \mathbb{R}^n$, there exists a neighborhood $\mathcal{U}$ of $x$ and $M > 0$ such that, for all $u \in \mathcal{U}$, for all $y \in D(u)$, $\|y\| < M$.*

**Definition 4 (Conservative set-valued field)** *A set-valued mapping $D : \mathbb{R}^n \rightrightarrows \mathbb{R}^m$ is a conservative field if the following conditions hold:*

    *1. For all $x \in \mathbb{R}^n$, $D(x)$ is nonempty.*

2. *D has a closed graph and is locally bounded.*

3. *For any absolutely continuous curve $\gamma : [0,1] \to \mathbb{R}^n$ with $\gamma(0) = \gamma(1)$,*

$$\int_0^1 \max_{z \in D(\gamma(t))} \langle \dot\gamma(t), z \rangle dt = 0.$$

Although conservative fields are not assumed to be locally bounded in [14], we add this restriction here to ensure they are upper semicontinuous. This will allow us to use a nonsmooth Lyapunov method [8] to prove convergence of first-order algorithms.

**Definition 5 (Monotone operator)** *A set-valued mapping $D : \mathbb{R}^n \rightrightarrows \mathbb{R}^m$ is called a monotone operator if, for all $x, y \in \mathbb{R}^n$, $u \in D(x)$, and $v \in D(y)$,*

$$\langle x - y, u - v \rangle \geq 0.$$

## A.2   A simpler and more operational view on definability

We recall basic definitions and results on definable sets and functions used in this work. More details on this theory can be found in [55, 23].

*We make a specific attempt to provide a new simple view on this subject by using dictionaries, in the hope that machine learning users consider utilizing these wonderful tools.*

The archetypal o-minimal structure is the collection of *semialgebraic* sets. Recall that a set $A \subset \mathbb{R}^n$ is semialgebraic if it can be written as

$$A = \bigcup_{i=1}^I \bigcap_{j=1}^J \{x \in \mathbb{R}^n : P_{ij}(x) < 0, \ Q_{ij}(x) = 0\}$$

where, for $i \in \{1, ..., I\}$ and $j \in \{1, ..., J\}$, $P_{ij}$ and $Q_{ij}$ are polynomials. The stability properties of semialgebraic sets may be axiomatized [51, 55] to give rise to the general notion of an o-minimal structure:

**Definition 6 (o-minimal structure)** *Let $\mathcal{O} = (\mathcal{O}_p)_{p \in \mathbb{N}}$ be a collection of sets such that, for all $p \in \mathbb{N}$, $\mathcal{O}_p$ is a set of subsets of $\mathbb{R}^p$. $\mathcal{O}$ is an o-minimal structure on $(\mathbb{R}, +, \cdot)$ if it satisfies the following axioms:*

1. *For all $p \in \mathbb{N}$, $\mathcal{O}_p$ is stable by finite intersection and union, complementation, and contains $\mathbb{R}^p$.*

2. *If $A \in \mathcal{O}_p$ then $A \times \mathbb{R}$ and $\mathbb{R} \times A$ belong to $\mathcal{O}_{p+1}$.*

3. *Denoting by $\pi$ the projection on the $p$ first coordinates, if $A \in \mathcal{O}_{p+1}$ then $\pi(A) \in \mathcal{O}_p$.*

4. *For all $p \in \mathbb{N}$, $\mathcal{O}_p$ contains the algebraic subsets of $\mathbb{R}^p$, i.e., sets of the form $\{x \in \mathbb{R}^p : P(x) = 0\}$, where $P : \mathbb{R}^p \to \mathbb{R}$ is a polynomial function.*

5. *The elements of $\mathcal{O}_1$ are exactly the finite unions of intervals.*

A subset $A \subset \mathbb{R}^n$ is said to be definable in an o-minimal structure $\mathcal{O} = (\mathcal{O}_p)_{p \in \mathbb{N}}$ if $\mathcal{O}_n$ contains $A$. A function $f : \mathbb{R}^n \to \mathbb{R}^m$ is said to be definable if its graph, a subset of $\mathbb{R}^{n+m}$, is definable.

Note that the collection of semialgebraic sets verifies 3 in Definition 6 according to the Tarski-Seidenberg theorem.

There are several major structures which have been explored [57, 55, 27]. But rather than relying on traditional description of these structures, we provide instead classes of functions that are contained in an o-minimal structure. The goals achieved are twofold:

- The classes we provide are o-minimal and thus all the results provided in the main text apply to functions in these classes.

- It is very easy to verify that a function belongs to one of the classes. Everything boils down to checking that the problem under consideration can be expressed in one of the dictionaries we provide.

Note however that we do not aim at providing neither a comprehensive nor a sharp picture of what could be done with o-minimal structures.

We consider first a collection of functions which will serve to establish dictionaries:

(a) Analytic functions restricted to semialgebraic compact domains (contained in their natural open domain), examples are $\cos$ and $\sin$ restricted to compact intervals.

(b) "Globally subanalytic functions": $\arctan, \tan_{]-\pi/2,\pi/2[}$ or any functions in (a) (see [27] for a precise definition of global subanalyticity).

(c) The $\log$ and $\exp$ functions.

(d) Functions of the form $x \mapsto x^r$ with $r$ a real constant and $x$ a positive real number. These can be represented as $x \mapsto \exp(r \log(x))$ which is definable in $(\mathbb{R}, \exp)$.

(e) Implicitly defined semialgebraic functions. That is, functions $G : \Omega \to \mathbb{R}^m$, with $\Omega$ open, which are maximal solutions (i.e., the domain $\Omega$ cannot be chosen to be bigger) to nonlinear equations of the type
$$F(x, G(x)) = 0$$
where $F$ is a semialgebraic function.

With this collection of functions we may build *elementary dictionaries*. To demonstrate, we consider the following dictionaries

$$\text{Dic(a)} = \{\text{functions satisfying (a)}\}$$
$$\text{Dic(d, e)} = \{\text{functions satisfying (d) or (e)}\}$$
$$\text{Dic(a, b, c, d, e)} = \{\text{functions satisfying (a) or (b) or (c) or (d) or (e)}\}$$

The last dictionary describes a larger class of functions, we shall come back on this later on.

Consider the dictionary $\mathcal{D} = \text{Dic}(\cdot)$ based on the properties (a)-(e) described above.

Then, in the spirit of [15], we can extend the idea of piecewise selection functions with the following three definitions.

**Definition 7 (Elementary $\mathcal{D}$-function)** *An elementary $\mathcal{D}$-function is a $C^2$ function described by a finite compositional expression involving the basic operations $\times, +, /$, multiplication by a constant, and the functions of $\mathcal{D}$ inside their domain of definition.*

Any elementary $\mathcal{D}$-function is definable in $\mathbb{R}_{\text{an,exp}}$ by stability of definable functions by composition. We shall denote $\mathfrak{S}\mathcal{D}$ the set of elementary $\mathcal{D}$-functions. For instance, the following functions belong to $\mathfrak{S}\mathcal{D}$:

- $x \mapsto \frac{1}{1+\exp(-x)}$.
- $x \mapsto \log(1 + \exp(x))$.
- $(\beta, \lambda) \mapsto \|X\beta - Y\|_2 + e^\lambda \|\beta\|_1$.

**Definition 8 (Elementary $\mathcal{D}$-index)** *Consider $r \in \mathbb{N}^*$, and $s : \mathbb{R}^n \to \{1, \ldots, r\}$. Then $s$ is said to be an elementary $\mathcal{D}$-index if, for $i \in \{1, \ldots, r\}$, each of the pre-images $s^{-1}(i)$ (i.e., the points in $\mathbb{R}^n$ such that $s$ selects the index $i$) can be written as*
$$\bigcup_{i=1}^{I} \bigcap_{j=1}^{J} \{x \in \mathbb{R}^n : g_{ij}(x) < 0, \ h_{ij}(x) = 0\}$$
*where, for $i \in \{1, \ldots, I\}$ and $j \in \{1, \ldots, J\}$, the $g_{ij}$ and $h_{ij}$ are elementary $\mathcal{D}$-functions.*

**Definition 9 (Piecewise $\mathcal{D}$-function)** *A function $f : \mathbb{R}^n \to \mathbb{R}^m$ is a piecewise $\mathcal{D}$-function if there exist $r \in \mathbb{N}^*$, elementary $\mathcal{D}$-functions $f_1, \ldots, f_r$, and an elementary $\mathcal{D}$-index $s : \mathbb{R}^n \to \{1, \ldots, r\}$ such that for all $x \in \mathbb{R}^n$,*
$$f(x) = f_{s(x)}(x).$$

We denote $\mathcal{PD}$ the set of piecewise $\mathcal{D}$-functions. With the assumptions we have on the dictionary $\mathcal{D}$, the piecewise selections we consider are all definable (it 's not always the case in general). Notice that piecewise log-exp functions [15] are a specific case of $\mathcal{D}$-functions with the dictionary $\mathcal{D} = \mathrm{Dic}(c) = \{\log, \exp\}$. It is easy to see that the following functions are in $\mathcal{PD}$ and thus definable:

- $x \mapsto \max(0, x)$ (relu).
- $x \mapsto \max(x_1, ..., x_n)$.
- sort function.
- $x \mapsto \begin{cases} \frac{1}{2}x^2 & \text{for } |x| \leq \delta, \\ \delta(|x| - \frac{1}{2}\delta), & \text{otherwise,} \end{cases}$ with $\delta > 0$ (Huber loss).

Moreover, composition of functions from $\mathcal{PD}$ are definable. This allows to say that if $\rho(w, x)$ is the output of a neural network built with usual elementary blocks (for instance Dense, Max Pooling or Conv layers), or even implicit layers involving functions in $\mathcal{PD}$, with input $x$ and weights $w$, then the empirical risk $\frac{1}{N}\sum_{i=1}^{N}\ell(\rho(w, x_i), y_i)$ is definable with respect to $w$ provided that $\ell$ is also in $\mathcal{PD}$.

**Remark 3** (a) (Small and big dictionaries) It may be puzzling for the reader to see that there is a dictionary that contains all the others. A major comment is in order: bigger is not always better. The bigger the dictionary is, the weaker some properties are. For instance, any piecewise selection $f : \mathbb{R}^n \to \mathbb{R}$ built upon $\mathrm{Dic}(a, b)$ satisfies $\|f(x)\| \leq c\|x\|^N$ for some $c > 0, N > 0$, which may have consequences in terms of convergence rates, see e.g., [5]. Thus in practice using the smallest dictionary possible may lead to sharper results. On top of this, there are no universal dictionaries [27].
(b) (PAP functions and definability) Recently PAP functions were introduced in order to deal with automatic differentiation matters [40]. To deal with such types of functions in our framework and have guarantees in terms of automatic differentiation, implicit differentiation or convergence properties, we need to view them through the dictionary paradigm. For this we consider the dictionary of analytic functions defined on $\mathbb{R}^p$ for some $p$. In that case, piecewise functions are not necessarily definable but their restrictions to any ball (or any compact semialgebraic subset) are definable.

# B    Results from Section 2

**Theorem 1 (The Clarke Jacobian is a minimal conservative Jacobian)** *Given a nonempty open subset $\mathcal{U}$ of $\mathbb{R}^n$ and $F : \mathcal{U} \subset \mathbb{R}^n \to \mathbb{R}^m$ locally Lipschitz, let $J_F$ be a convex-valued conservative Jacobian for F. Then for almost all $x \in \mathcal{U}$, $J_F(x) = \{\mathrm{Jac}\,F\}$ and for all $x \in \mathcal{U}$, $\mathrm{Jac}^{\,c}F(x) \subset J_F(x)$.*

**Proof:** Using [14, Lemma 4] for $i \in \{1, \ldots, m\}$, $[J_F]_i$ is a conservative map for $F_i$ on $\mathcal{U}$ and it is equal to $\nabla F_i$ on a set of full measure $S_i \subset \mathcal{U}$. Hence for all $x \in S := \bigcap_{i=1}^{m} S_i$, which is of full measure in $\mathcal{U}$, $J_F(x) = \mathrm{Jac}\,F(x)$. Since $S$ has full measure within $\mathcal{U}$, [56] gives the representation

$$\mathrm{Jac}^{\,c}F(x) = \mathrm{conv}\left\{\lim_{k \to +\infty} \mathrm{Jac}\,F(x_k) : x_k \in S, x_k \underset{k \to +\infty}{\to} x\right\}, \text{ for any } x \in \mathcal{U}.$$

But since $J_F$ coincides with $\mathrm{Jac}\,F$ throughout $S$, we have

$$\mathrm{Jac}^{\,c}F(x) = \mathrm{conv}\left\{\lim_{k \to +\infty} J_F(x_k) : x_k \in S, x_k \underset{k \to +\infty}{\to} x\right\}$$

for each $x \in \mathcal{U}$. Finally, by graph closedness and convexity of $J_F$ we get, for each $x \in \mathcal{U}$,

$$\mathrm{Jac}^{\,c}F(x) \subset \mathrm{conv}\left\{J_F\left(\lim_{k \to +\infty} x_k\right) : x_k \in S, x_k \underset{k \to +\infty}{\to} x\right\} = J_F(x).$$

$\square$

**Proposition 1 (Decomposition of conservative fields)** *Let $J_F$ be a conservative Jacobian for F, then there is a residual $R$ such that*

$$J_F \subset \mathrm{Jac}^{\,c}F + R.$$

**Proof:** We have obviously the inclusion

$$J_F \subset \operatorname{Jac}{}^c F + (J_F - \operatorname{Jac}{}^c F),$$

so it suffices to remark that $(J_F - \operatorname{Jac}{}^c F)$ is residual due to the conservativity properties of both $J_F$ and $\operatorname{Jac}{}^c F$. $\qquad\square$

**Theorem 2 (Implicit differentiation)** *Let $F : \mathbb{R}^n \times \mathbb{R}^m \to \mathbb{R}^m$ be path differentiable on $\mathcal{U} \times \mathcal{V} \subset \mathbb{R}^n \times \mathbb{R}^m$ an open set and $G : \mathcal{U} \to \mathcal{V}$ a locally Lipschitz function such that, for each $x \in \mathcal{U}$,*

$$F(x, G(x)) = 0. \tag{16}$$

*Furthermore, assume that for each $x \in \mathcal{U}$, for each $[A\ B] \in J_F(x, G(x))$, the matrix $B$ is invertible where $J_F$ is a conservative Jacobian for $F$. Then, $G : \mathcal{U} \to \mathcal{V}$ is path differentiable with conservative Jacobian given, for each $x \in \mathcal{U}$, by*

$$J_G \colon x \rightrightarrows \left\{ -B^{-1}A : [A\ B] \in J_F(x, G(x)) \right\}.$$

**Proof:** Let $\gamma : [0, 1] \to \mathcal{U}$ be absolutely continuous, then the composition $G \circ \gamma$ is also absolutely continuous since $G$ is locally Lipschitz. By (16) we have, for all $t \in [0, 1]$,

$$F(\gamma(t), G(t)) = 0$$

which we can differentiate almost everywhere; for almost every $t \in [0, 1]$, for any $[A\ B] \in J_F(\gamma(t), G(\gamma(t)))$,

$$[A\ B] \begin{bmatrix} \dot{\gamma}(t) \\ \frac{\mathrm{d}}{\mathrm{d}t} G(\gamma(t)) \end{bmatrix} = 0 \implies -A\dot{\gamma}(t) = B \frac{\mathrm{d}}{\mathrm{d}t} G(\gamma(t)).$$

Since $B$ is assumed to be invertible, we have, for almost every $t \in [0, 1]$,

$$-B^{-1}A\dot{\gamma}(t) = \frac{\mathrm{d}}{\mathrm{d}t} G(\gamma(t)).$$

The set-valued mapping $J_G \colon x \rightrightarrows \left\{ -B^{-1}A : [A\ B] \in J_F(x, G(x)) \right\}$ is nonempty, locally bounded, and has a closed graph for each $x \in \mathcal{U}$ since $J_F(x, G(x))$ is a conservative Jacobian and $B$ is invertible . We conclude that $G$ is path differentiable on $\mathcal{U}$ with conservative Jacobian $J_G$. $\qquad\square$

**Corollary 1 (Path differentiable implicit function theorem)** *Let $F : \mathbb{R}^n \times \mathbb{R}^m \to \mathbb{R}^m$ be path differentiable with conservative Jacobian $J_F$. Let $(\hat{x}, \hat{y}) \in \mathbb{R}^n \times \mathbb{R}^m$ be such that $F(\hat{x}, \hat{y}) = 0$. Assume that $J_F(\hat{x}, \hat{y})$ is convex and that, for each $[A\ B] \in J_F(\hat{x}, \hat{y})$, the matrix $B$ is invertible. Then, there exists an open neighborhood $\mathcal{U} \times \mathcal{V} \subset \mathbb{R}^n \times \mathbb{R}^m$ of $(\hat{x}, \hat{y})$ and a path differentiable function $G : \mathcal{U} \to \mathcal{V}$ such that the conclusion of Theorem 2 holds.*

**Proof:** Since $J_F(\hat{x}, \hat{y})$ is convex, it follows from Theorem 1 that $\operatorname{Jac}{}^c F(\hat{x}, \hat{y}) \subset J_F(\hat{x}, \hat{y})$ and thus, for any $[A\ B] \in \operatorname{Jac}{}^c F(\hat{x}, \hat{y})$, $B$ is invertible, i.e., the conditions to apply [20, 7.1 Corollary] to $F$ are satisfied. Therefore there exists an open neighborhood $\mathcal{U}_1 \times \mathcal{V}_1 \subset \mathbb{R}^n \times \mathbb{R}^m$ of $(\hat{x}, \hat{y})$ and a locally Lipschitz function $G : \mathcal{U}_1 \to \mathcal{V}_1$ such that, for all $x \in \mathcal{U}_1$,

$$F(x, G(x)) = 0.$$

By the continuity of the determinant and the fact that $J_F$ has a closed graph, there exists an open neighborhood $\mathcal{U}_2 \times \mathcal{V}_2 \subset \mathbb{R}^n \times \mathbb{R}^m$ of $(\hat{x}, \hat{y})$ such that, for all $(x, y) \in \mathcal{U}_2 \times \mathcal{V}_2$, for all $[A\ B] \in J_F(x, y)$, the matrix $B$ is invertible. Let $\mathcal{U} \times \mathcal{V} := (\mathcal{U}_1 \cap \mathcal{U}_2) \times (\mathcal{V}_1 \cap \mathcal{V}_2)$, which is an open neighborhood of $(\hat{x}, \hat{y})$. Then the requirements of Theorem 2 are met for $F$, $J_F$, and $G$ on $\mathcal{U} \times \mathcal{V}$ and the desired claims follow. $\qquad\square$

**Corollary 2 (Path differentiable inverse function theorem)** *Let $\mathcal{U}$ and $\mathcal{V}$ be open neighborhoods of $0$ in $\mathbb{R}^n$ and $\Phi : \mathcal{U} \to \mathcal{V}$ path differentiable with $\Phi(0) = 0$. Assume that $\Phi$ has a conservative Jacobian $J_\Phi$ such that $J_\Phi(0)$ contains only invertible matrices. Then, locally, $\Phi$ has a path differentiable inverse $\Psi$ with a conservative Jacobian given by*

$$J_\Psi(y) = \left\{ A^{-1} : A \in J_\Phi(\Psi(y)) \right\}.$$

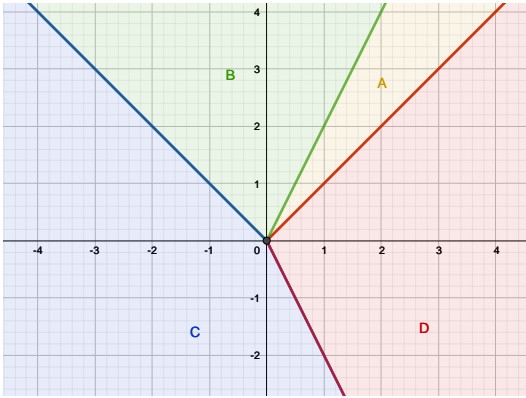

Figure 4: Illustration of the four different sets in the explicit piecewise affine representation of $\Psi = \Phi^{-1}$.

**Proof:** Consider the function $F(x, y) = x - \Phi(y)$ and observe that it satisfies the assumptions of Corollary 1, so that we obtain a function $G$ which is exactly the desired inverse. $\square$

It is tempting to think that Corollary 2 should come with a formula of the type

$$\mathrm{Jac}^c\,\Psi(z) = [\mathrm{Jac}^c\,\Phi(\Psi(z))]^{-1},$$

for all $z$ in a neighborhood of 0. This happens to be false, making the use of the notion of conservativity necessary to catpure the artifacts resulting from application of ordinary calculus rules to nonsmooth inverse functions. Note that since the inverse function theorem is a special case of the implicit function theorem, this also rules out a Clarke calculus for implicit functions.

**Example 1 (Counterexample to a potential "Clarke implicit differential calculus")** We follow the example given by Clarke [20, Remark 7.1.2]. Consider the mapping $\Phi : \mathbb{R}^2 \to \mathbb{R}^2$ given by

$$\Phi(x, y) = (|x| + y, 2x + |y|).$$

It is locally Lipschitz and semialgebraic and thus path differentiable with its Clarke Jacobian a conservative Jacobian. We have the following explicit piecewise linear representation

$$\Phi(x, y) = \begin{cases} (x + y, 2x + y) & \text{if } x \geq 0 \text{ and } y \geq 0, \\ (x + y, 2x - y) & \text{if } x \geq 0 \text{ and } y \leq 0, \\ (-x + y, 2x - y) & \text{if } x \leq 0 \text{ and } y \leq 0, \\ (-x + y, 2x + y) & \text{if } x \leq 0 \text{ and } y \geq 0 \end{cases}$$

from which we deduce that the Clarke Jacobian of $\Phi$ has the following structure

$$\mathrm{Jac}^c\,\Phi(0) = \mathrm{conv}\left\{ \begin{bmatrix} 1 & 1 \\ 2 & 1 \end{bmatrix}, \begin{bmatrix} 1 & 1 \\ 2 & -1 \end{bmatrix}, \begin{bmatrix} -1 & 1 \\ 2 & -1 \end{bmatrix}, \begin{bmatrix} -1 & 1 \\ 2 & 1 \end{bmatrix} \right\}$$

where the matrices correspond to linear maps in the explicit definition of $\Phi$. Therefore $\mathrm{Jac}^c\,\Phi(0)$ is an affine set whose dimension is 2. In addition, it contains only invertible matrices [20, Remark 7.1.2]. We will use the following explicit matrix inverses:

$$\begin{bmatrix} 1 & 1 \\ 2 & 1 \end{bmatrix}^{-1} = \begin{bmatrix} -1 & 1 \\ 2 & -1 \end{bmatrix}, \quad \begin{bmatrix} 1 & 1 \\ 2 & -1 \end{bmatrix}^{-1} = \frac{1}{3}\begin{bmatrix} 1 & 1 \\ 2 & -1 \end{bmatrix}, \quad \begin{bmatrix} -1 & 1 \\ 2 & 1 \end{bmatrix}^{-1} = \frac{1}{3}\begin{bmatrix} -1 & 1 \\ 2 & 1 \end{bmatrix}.$$

Using the above, one can verify that $\Phi$ is a homeomorphism whose inverse is also piecewise linear. We set $\Psi = \Phi^{-1}$; it is given by

$$\Psi(u, v) = (v - u, 2u - v) \quad \text{for } (u, v) \in A,$$

$$\Psi(u, v) = \frac{1}{3}(u + v, 2u - v) \quad \text{for } (u, v) \in B,$$

$$\Psi(u, v) = (u + v, 2u + v) \quad \text{for } (u, v) \in C,$$

$$\Psi(u, v) = \frac{1}{3}(v - u, 2u + v) \quad \text{for } (u, v) \in D,$$

where the subsets $A, B, C, D$ form a "partition"[2] of $\mathbb{R}^2$

$$
\begin{aligned}
A &= \left\{ (u,v) \in \mathbb{R}^2 : v - u \geq 0, 2u - v \geq 0 \right\} & \text{(corresponding to } x \geq 0, y \geq 0\text{)}, \\
B &= \left\{ (u,v) \in \mathbb{R}^2 : u + v \geq 0, 2u - v \leq 0 \right\} & \text{(corresponding to } x \geq 0, y \leq 0\text{)}, \\
C &= \left\{ (u,v) \in \mathbb{R}^2 : u + v \leq 0, 2u + v \leq 0 \right\} & \text{(corresponding to } x \leq 0, y \leq 0\text{)}, \\
D &= \left\{ (u,v) \in \mathbb{R}^2 : v - u \leq 0, 2u + v \geq 0 \right\} & \text{(corresponding to } x \leq 0, y \geq 0\text{)}.
\end{aligned}
$$

A graphical representation of these sets is given in Figure 4.

From this explicit piecewise linear representation of $\Psi$, we deduce that its Clarke Jacobian at $0$ is the following

$$
\mathrm{Jac}^c \, \Psi(0) = \mathrm{conv} \left\{ \begin{bmatrix} 1 & 1 \\ 2 & 1 \end{bmatrix}, \begin{bmatrix} -1 & 1 \\ 2 & -1 \end{bmatrix}, \frac{1}{3} \begin{bmatrix} -1 & 1 \\ 2 & 1 \end{bmatrix}, \frac{1}{3} \begin{bmatrix} 1 & 1 \\ 2 & -1 \end{bmatrix} \right\}.
$$

For a given subset of linear space we denote by $\mathrm{aff}\, F$ the affine span of $F$. It is easy to see that $\dim \mathrm{aff}[\, \mathrm{Jac}^c \, \Phi(0)] = 2$ while $\dim \mathrm{aff}\, [\mathrm{Jac}^c \, \Psi(0)] = 3$. More concretely, vectorialize the set $\mathrm{Jac}^c \, \Psi(0)$ at $M = \frac{1}{3} \begin{bmatrix} 1 & 1 \\ 2 & -1 \end{bmatrix}$ by considering the matrices given by

$$
\begin{bmatrix} 1 & 1 \\ 2 & 1 \end{bmatrix} - M, \quad \begin{bmatrix} -1 & 1 \\ 2 & -1 \end{bmatrix} - M, \quad \frac{1}{3} \begin{bmatrix} -1 & 1 \\ 2 & 1 \end{bmatrix} - M
$$

that is

$$
\frac{1}{3} \begin{bmatrix} 2 & 2 \\ 4 & 4 \end{bmatrix}, \quad \frac{1}{3} \begin{bmatrix} -4 & 2 \\ 4 & -2 \end{bmatrix}, \quad \frac{1}{3} \begin{bmatrix} -2 & 0 \\ 0 & 2 \end{bmatrix}.
$$

These matrices are independent so that $\mathrm{Jac}^c \, \Psi(0)$ is an affine set whose dimension is $3$.

Matrix inversion is a semialgebraic diffeomorphism (when restricted to invertible matrices) so it preserves dimension. For this reason the set $[\mathrm{Jac}^c \, \Psi(0)]^{-1} = \{M^{-1}, M \in \mathrm{Jac}^c \, \Psi(0)\}$ is a semialgebraic set of dimension $3$, and we have

$$
[\mathrm{Jac}^c \, \Psi(0)]^{-1} \not\subset [\mathrm{Jac}^c \, \Phi(0)]. \tag{17}
$$

However, we have shown that $z \mapsto [\mathrm{Jac}^c \, \Psi(\Phi(z))]^{-1}$ is a conservative Jacobian. This example excludes the possibility of a simple inverse (implicit) function theorem with a "Clarke Jacobian calculus" and illustrates the requirement for a more flexible notion (conservativity) when using calculus rules in an implicit function (or inverse function) context.

**The Lipschitz definable implicit and inverse function theorems.** In the definable (e.g. semialgebraic case) our results have a remarkably simple expression that we give below.

**Theorem 4 (Lipschitz definable inverse function theorem)** *Let $\mathcal{U}$ and $\mathcal{V}$ be two open neighborhoods of $0$ in $\mathbb{R}^n$ and $\Phi : \mathcal{U} \to \mathcal{V}$ a locally Lipschitz definable mapping with $\Phi(0) = 0$. Assume that $\Phi$ has a conservative Jacobian $J_\Phi$ such that $J_\Phi(0)$ contains only invertible matrices. Then, locally, $\Phi$ has locally Lipschitz definable inverse $\Psi$ with a conservative Jacobian given by*

$$
J_\Psi(y) = \left\{ A^{-1} : A \in J_\Phi(\Psi(y)) \right\}.
$$

**Proof:** It suffices to use the fact that definable mappings are path differentiable, see [14], and that the the graph of $\Psi$ is given by a first-order formula. $\square$

The same type of arguments gives:

**Theorem 5 (Lipschitz definable implicit function theorem)** *Let $F : \mathbb{R}^n \times \mathbb{R}^m \to \mathbb{R}^m$ be locally Lipchitz and definable with conservative Jacobian $J_F$. Let $(\hat{x}, \hat{y}) \in \mathbb{R}^n \times \mathbb{R}^m$ be such that $F(\hat{x}, \hat{y}) = 0$. Assume that $J_F(\hat{x}, \hat{y})$ is convex and that, for each $[A\ B] \in J_F(\hat{x}, \hat{y})$, the matrix $B$ is invertible. Then, there exists an open neighborhood $\mathcal{U} \times \mathcal{V} \subset \mathbb{R}^n \times \mathbb{R}^m$ of $(\hat{x}, \hat{y})$ and a locally Lipschitz definable function $G : \mathcal{U} \to \mathcal{V}$ such that, for all $x \in \mathcal{U}$,*

$$
F(x, G(x)) = 0.
$$

*Moreover, for each $x \in \mathcal{U}$, the mapping $J_G : x \rightrightarrows \{-B^{-1}A : [A\ B] \in J_F(x, G(x))\}$ is conservative for $G$.*

---

[2]Each piece having two half lines in common with other pieces.

# C Results from Section 3

## C.1 Monotone operator deep equilibrium networks

**Proposition 3 (Path differentiation through monotone layers)** *Assume that $J_\sigma$ is convex-valued and that, for all $J \in J_\sigma(Wz(W,b) + b)$, the matrix $(\mathrm{Id}_m - JW)$ is invertible. Consider a loss-like function $\ell \colon \mathbb{R}^m \to \mathbb{R}$ with conservative gradient $D_\ell \colon \mathbb{R}^m \rightrightarrows \mathbb{R}^m$, then $g \colon (W,z) \mapsto \ell(z(W,b))$ is path differentiable and has a conservative gradient $D_g$ defined through*

$$D_g \colon (W,b) \rightrightarrows \left\{ J^T(\mathrm{Id}_m - JW)^{-T}vz^T, J^T(\mathrm{Id}_m - JW)^{-T}v) : J \in J_\sigma(Wz+b), v \in D_\ell(z) \right\}.$$

**Proof:** The quantity $z(W,b)$ is defined implicitly by the relation

$$z(W,b) - \sigma(Wz(W,b) + b) = 0. \tag{18}$$

We set $M = m + m + m \times m$ and represent the pair $(W,b) \in \mathbb{R}^{m \times m} \times \mathbb{R}^m$ as $(w_1, \dots, w_m, b) \in \mathbb{R}^{M-m}$ where $w_i \in \mathbb{R}^m$ is the $i$-th row of $W$ for $i \in \{1, \dots, m\}$. We denote by $\mathcal{B} \colon \mathbb{R}^M \to \mathbb{R}^m$ the bilinear map defined as

$$\mathcal{B}(w_1, \dots, w_m, b, z) := Wz + b$$

so that $\mathcal{B}$ is infinitely differentiable. Equation (18) is then equivalent to

$$z - (\sigma \circ \mathcal{B})(w_1, \dots, w_m, b, z) = 0.$$

We denote by $F$ the mapping

$$F \colon (w_1, \dots, w_m, b, z) \mapsto z - (\sigma \circ \mathcal{B})(w_1, \dots, w_m, b, z).$$

For $i \in \{1 \dots m\}$, denote by $Z_i \in \mathbb{R}^{m \times m}$ the matrix whose $i$-th row is $z$, and remaining rows are null. The Jacobian of $\mathcal{B}$, $\mathrm{Jac}\,\mathcal{B} \colon \mathbb{R}^M \to \mathbb{R}^{m \times M}$ is as follows:

$$\mathrm{Jac}\,\mathcal{B}(w_1, \dots, w_m, b, z) = [Z_1 \quad \dots \quad Z_m \quad \mathrm{Id}_m \quad W]$$

where $[A\,B]$ is used to denote the columnwise concatenation of matrices $A$ and $B$. By hypothesis, we have a conservative Jacobian for $\sigma$, $J_\sigma$. Conservative Jacobians may be composed as usual Jacobians [14, Lemma 5]. As $\mathcal{B}$ is continuously differentiable, $\mathrm{Jac}\,\mathcal{B}$ is also a conservative Jacobian for $\mathcal{B}$. Therefore, we have the following conservative Jacobian for $F$,

$$J_F(w_1, \dots, w_m, b, z) \rightrightarrows \left\{ [-JZ_1 \quad \dots \quad -JZ_m \quad -J \quad \mathrm{Id}_m - JW], J \in J_\sigma(Wz+b) \right\}.$$

Finally, by hypothesis, for any $W, b$, and $z$ such that $F(W,b,z) = 0$ and any $J \in J_\sigma(Wz+b)$, the matrix $\mathrm{Id}_m - JW$ is invertible. Therefore, Theorem 2 applies and, setting $\tilde{M} = m \times m + m = M - m$, the set-valued mapping

$$J_z \colon \mathbb{R}^{\tilde{M}} \rightrightarrows \mathbb{R}^{m \times \tilde{M}}$$

$$(w_1, \dots, w_m, b) \rightrightarrows \left\{ (\mathrm{Id}_m - JW)^{-1}J\,[Z_1 \quad \dots \quad Z_m \quad \mathrm{Id}_m], J \in J_\sigma(Wz+b) \right\}$$

is conservative for $(W,b) \mapsto z(W,b)$ as defined in (18). We denote by $Z \in \mathbb{R}^{m \times \tilde{M}}$ the matrix $[Z_1 \dots Z_m \, \mathrm{Id}_m]$ appearing in the definition of $J_z$. Given the loss function $\ell$, the mapping $J_\ell \colon z \mapsto \{v^T, v \in D_\ell(z)\}$ is a conservative Jacobian for $\ell$ [14, Lemma 3] and therefore, the set-valued mapping

$$J_g \colon \mathbb{R}^{\tilde{M}} \rightrightarrows \mathbb{R}^{1 \times \tilde{M}}$$

$$(w_1, \dots, w_m, b) \rightrightarrows \left\{ v^T(\mathrm{Id}_m - JW)^{-1}JZ, J \in J_\sigma(Wz+b), v \in D_\ell(z(W,b)) \right\}$$

is a conservative Jacobian for $g \colon (W,b) \mapsto \ell(z(W,b))$. Using [14, Lemma 4], we obtain a conservative gradient field for $g$ by a simple transposition as follows

$$D_g \colon (w_1, \dots, w_m, b) \rightrightarrows \left\{ Z^T J^T(\mathrm{Id}_m - JW)^{-T}v, J \in J_\sigma(Wz+b), v \in D_\ell(z(W,b)) \right\}.$$

We now identify the terms by block computation; recall that $Z = [Z_1 \dots Z_m \, \mathrm{Id}_m]$ and that $Z_i \in \mathbb{R}^{m \times m}$ is the matrix whose $i$-th row is $z$ with remaining rows null for each $i \in \{1, \dots, m\}$. The term associated to $b$ corresponds to the last $m \times m$ block in $Z$, it is indeed of the form $J^T(\mathrm{Id}_m - JW)^{-T}v$. Similarly, for each $i \in \{1, \dots, m\}$, the term associated to $w_i$ is of the form $Z_i^T J^T(\mathrm{Id}_m - JW)^{-T}v$. For any $a \in \mathbb{R}^m$ and $i \in \{1, \dots, m\}$, we have $Z_i^T a = a_i z$ where $a_i$ is the $i$-th coordinate of $a$ and $z$ corresponds to the $i$-th row of $Z_i^T$. So the component associated to $w_i$ in $D_g$ is of the form $[J^T(\mathrm{Id}_m - JW)^{-T}v]_i z$, where $[\cdot]_i$ denotes the $i$-th coordinate. Since $w_i$ denotes the $i$-th row of $W$, rearranging this expression in matrix format provides a term of the form $J^T(\mathrm{Id}_m - JW)^{-T}vz^T$ for the $W$ component. This concludes the proof. $\square$

## C.2 Optimization layers: the conic program case

Let us first expand on the link between zeros of the residual map and KKT solutions. We provide a simplified view of [17, 3], ignoring cases of infeasibility and unboundedness. Note that this corresponds to enforcing $w = 1$ as done in [2, 3].

The following is due to Moreau [44]. Recall that the polar of a closed convex cone $\mathcal{K} \subset \mathbb{R}^m$ is given by $\mathcal{K}^\circ = \{x \in \mathbb{R}^m, \, y^T x \leq 0, \, \forall y \in \mathcal{K}\}$, in which case $(\mathcal{K}^\circ)^\circ = \mathcal{K}$ and the dual cone satisfies $\mathcal{K}^* = -\mathcal{K}^\circ$.

**Proposition 6** *Let $s, y, v \in \mathbb{R}^m$; the following are equivalent*

- $v = s + y$, $s \in \mathcal{K}$, $y \in \mathcal{K}^\circ$, $s^T y = 0$.

- $s = P_\mathcal{K}(v)$, $y = P_{\mathcal{K}^\circ}(v)$.

We may reformulate this equivalence as follows, using changes of signs on $y$ and $v$, noticing that $-P_{\mathcal{K}^\circ}(-\cdot) = P_{\mathcal{K}^*}(\cdot)$ since $\mathcal{K}^* = -\mathcal{K}^\circ$,

(i) $v = y - s$, $\quad s \in \mathcal{K}$, $\quad y \in \mathcal{K}^*$, $\quad s^T y = 0$.
(ii) $s = P_{\mathcal{K}^*}(v) - v$, $\quad y = P_{\mathcal{K}^*}(v)$.

Now the KKT system in $(x, y, s)$ for problem (P) and (D) can be written as follows (see, for example, [17]),

$$\begin{aligned} A^T y + c = 0, &\quad y \in \mathcal{K}^* \\ -Ax + b = s, &\quad s \in \mathcal{K} \\ s^T y = 0 & \end{aligned}$$

which is equivalent, by setting $v = y - s$ and $u = x$, to

$$\begin{aligned} A^T P_{\mathcal{K}^*}(v) + c = 0 \\ -Au + b = P_{\mathcal{K}^*}(v) - v \end{aligned} \tag{19}$$

The system (19) is equivalent to $\mathcal{N}(z, A, b, c) = 0$ with $z = (u, v)$. We have shown that $(x, y, s)$ is a KKT solution to the system if and only if $(x, y, s) = (u, P_{\mathcal{K}^*}(v), P_{\mathcal{K}^*}(v) - v) = \phi(z)$ for $z = (x, y - s)$ such that $\mathcal{N}(z, A, b, c) = 0$.

**Proposition 4 (Path differentiation through cone programming layers)** *Assume that $P_{\mathcal{K}^*}$, $\mathcal{N}$ are path differentiable, denote respectively by $J_{P_{\mathcal{K}^*}}$, $J_\mathcal{N}$ corresponding convex-valued conservative Jacobians. Assume that for all $A, b, c \in \mathbb{R}^{m \times n} \times \mathbb{R}^m \times \mathbb{R}^n$, $z = \nu(A, b, c) \in \mathbb{R}^n \times \mathbb{R}^m$ is the unique solution to $\mathcal{N}(z, A, b, c) = 0$, and that all matrices formed from the $N$ first columns of $J_\mathcal{N}(z, A, b, c)$ are invertible. Then, $\phi$, $\nu$, and* sol *are path differentiable functions with conservative Jacobians:*

$$J_\nu(A, b, c) := \left\{ -U^{-1} V : [U \ V] \in J_\mathcal{N}(\nu(A, b, c), A, b, c) \right\},$$

$$J_\phi(z) := \begin{bmatrix} \mathrm{Id}_n & 0 \\ 0 & J_{P_{\mathcal{K}^*}}(v) \\ 0 & (J_{P_{\mathcal{K}^*}}(v) - \mathrm{Id}_m) \end{bmatrix},$$

$$J_{\mathrm{sol}}(A, b, c) := J_\phi(\nu(A, b, c)) J_\nu(A, b, c).$$

**Proof:** First, the assumptions clearly ensure that $\nu$ and sol are single-valued and can be interpreted as functions such that $\mathrm{sol} = \phi \circ \nu$. By assumption, $\phi$ is differentiable. We will first use Corollary 1 to obtain a conservative Jacobian for $\nu$ and then justify the expression for $\phi$. The composition obtained for $J_{\mathrm{sol}}$ results from Proposition 2.

Let $A, b, c \in \mathbb{R}^{m \times n} \times \mathbb{R}^m \times \mathbb{R}^n$, $z := (u, v) \in \mathbb{R}^n \times \mathbb{R}^m$ such that $\mathcal{N}(z, A, b, c) = 0$. By assumption, the submatrices formed from the first $N$ columns of $J_\mathcal{N}(z, A, b, c)$ are invertible. Then applying Corollary 1, there exist open neighborhoods $\mathcal{U} \subset \mathbb{R}^{m \times n} \times \mathbb{R}^m \times \mathbb{R}^n$ and $\mathcal{V} \subset \mathbb{R}^N$ and a locally Lipschitz function $G : \mathcal{U} \to \mathcal{V}$ satisfying, for all $s \in \mathcal{U}$ $\mathcal{N}(G(s), s) = 0$ with $G$ is path differentiable. Since, by assumption, the solution $\nu(A, b, c)$ to $\mathcal{N}(\nu(A, b, c), A, b, c) = 0$ is unique, $\nu$ coincides with $G$ on $\mathcal{U}$. Thus, $\nu$ is path differentiable and a conservative Jacobian for $\nu$ is given by:

$$J_\nu(A, b, c) = \left\{ -U^{-1} V : [U \ V] \in J_\mathcal{N}(\nu(A, b, c), A, b, c) \right\}$$

Let us now turn to $\phi$. Since $P_{\mathcal{K}^*}$ has for conservative Jacobian $J_{P_{\mathcal{K}^*}}$, we may construct a conservative Jacobian for the function $\phi$ as follows using [14, Lemmas 3, 4, and 5]:

$$J_\phi(z) = \begin{bmatrix} \mathrm{Id}_n & 0 \\ 0 & J_{P_{\mathcal{K}^*}}(v) \\ 0 & (J_{P_{\mathcal{K}^*}}(v) - \mathrm{Id}_m) \end{bmatrix}.$$

It follows from Proposition 2 that the composition $\mathrm{sol} = \phi \circ \nu$ is also path differentiable with conservative Jacobian

$$J_{\mathrm{sol}}(A, b, c) = J_\phi(\nu(A, b, c)) J_\nu(A, b, c).$$

$\square$

### C.3 Hyperparameter selection for nonsmooth Lasso-type model

**Proposition 5 (Conservative Jacobian for the solution mapping)** *For all $\lambda \in \mathbb{R}$, assume $X_{\mathcal{E}}^T X_{\mathcal{E}}$ is invertible where $X_{\mathcal{E}}$ is the submatrix of $X$ formed by taking the columns indexed by $\mathcal{E}$. Then $\hat{\beta}(\lambda)$ is single-valued, path differentiable with conservative Jacobian, $J_{\hat{\beta}}(\lambda)$, given for all $\lambda$ as*

$$\left\{ \left[ -e^\lambda \left( \mathrm{Id}_p - \mathrm{diag}\,(q) \left( \mathrm{Id}_p - X^T X \right) \right)^{-1} \mathrm{diag}\,(q)\,\mathrm{sign}\left( \hat{\beta} - X^T \left( X\hat{\beta} - y \right) \right) \right] : q \in \mathcal{M}(\lambda) \right\}$$

*where $\mathcal{M}(\lambda) \subset \mathbb{R}^p$ is the set of vectors $q$ such that $q_i \in \begin{cases} \{1\} & i \in \mathrm{supp}\,\hat{\beta} \\ [0, 1] & i \in \mathcal{E} \setminus \mathrm{supp}\,\hat{\beta} \\ \{0\} & i \notin \mathcal{E} \end{cases}$.*

**Proof:** Our goal is to apply Corollary 1 to the path differentiable "optimality gap" function $F : \mathbb{R} \times \mathbb{R}^p \to \mathbb{R}^p$ defined in (3). For each $\lambda \in \mathbb{R}$, the invertibility of $X_{\mathcal{E}}^T X_{\mathcal{E}}$ guarantees the uniqueness of $\hat{\beta}(\lambda)$ (see [46], [42, Lemma 1]), i.e., $\hat{\beta} : \mathbb{R} \to \mathbb{R}^p$ is a function. Because $\|\cdot\|_1$ is separable, the components of the prox can be written, for any $(\lambda, u) \in \mathbb{R} \times \mathbb{R}^p$, for all $i \in \{1, \dots, p\}$, as

$$[\mathrm{prox}_{e^\lambda \|\cdot\|_1}(u)]_i = \mathrm{prox}_{e^\lambda |\cdot|}(u_i)$$

which have Clarke subdifferentials

$$\partial^c \mathrm{prox}_{e^\lambda |\cdot|} : u_i \rightrightarrows \mathbb{1}_{u_i, e^\lambda} \quad \times \begin{bmatrix} 1 \\ -\mathrm{sign}(u_i) \end{bmatrix} \quad \text{where} \quad \mathbb{1}_{e^\lambda}(u_i) := \begin{cases} 0 & |u_i| < e^\lambda \\ [0, 1] & |u_i| = e^\lambda \\ 1 & |u_i| > e^\lambda \end{cases}.$$

Thus a conservative Jacobian for $F$ at $(\lambda, \beta)$ is given by

$$J_F : (\lambda, \beta) \rightrightarrows \{ [\underbrace{e^\lambda \mathrm{diag}(q) \mathrm{sign}(\beta - X^T(X\beta - y))}_A \quad \underbrace{\mathrm{Id}_p - \mathrm{diag}(q) \left( \mathrm{Id}_p - X^T X \right)}_B] : q \in \mathcal{C} \}$$

$$(20)$$

with $\mathcal{C} := \{ q : q_i \in \mathbb{1}_{e^\lambda}(\beta_i - X_i^T(X\beta - y)) \}$. Let us estimate the factors $q_i$ above in terms of the equicorrelation set $\mathcal{E}$. Recall the KKT conditions [54] for the Lasso problem; a solution $\hat{\beta}$ must satisfy

$$X^T \left( y - X\hat{\beta} \right) = e^\lambda \delta \qquad \text{where} \qquad \delta_i \in \begin{cases} \left\{ \mathrm{sign}\left( \hat{\beta}_i \right) \right\} & i \in \mathrm{supp}\,\hat{\beta} \\ [-1, 1] & i \notin \mathrm{supp}\,\hat{\beta} \end{cases}. \qquad (21)$$

For $i \in \mathrm{supp}\,\hat{\beta}$, (21) gives

$$X_i^T \left( y - X\hat{\beta} \right) = e^\lambda \mathrm{sign}\left( \hat{\beta}_i \right) \implies \mathrm{sign}\left( X_i^T \left( y - X\hat{\beta} \right) \right) = \mathrm{sign}\left( \hat{\beta}_i \right)$$

$$\implies \mathrm{sign}\left( \hat{\beta}_i \right) = \mathrm{sign}\left( \hat{\beta}_i - X_i^T \left( X\hat{\beta} - y \right) \right)$$

$$= \mathrm{sign}\left( X_i^T \left( y - X\hat{\beta} \right) \right).$$

Noting that $\left|\hat{\beta}_i\right| > 0$ and $\left|X_i^T\left(y - X\hat{\beta}\right)\right| = e^\lambda$ since $i \in \operatorname{supp}\hat{\beta} \subset \mathcal{E}$,

$$
\begin{aligned}
\left|\hat{\beta}_i - X_i^T\left(X\hat{\beta} - y\right)\right| &= \operatorname{sign}\left(\hat{\beta}_i - X_i^T\left(X\hat{\beta} - y\right)\right)\left(\hat{\beta}_i - X_i^T\left(X\hat{\beta} - y\right)\right) \\
&= \operatorname{sign}\left(\hat{\beta}_i\right)\hat{\beta}_i + \operatorname{sign}\left(X_i^T\left(y - X\hat{\beta}\right)\right)X_i^T\left(y - X\hat{\beta}\right) \\
&= \underbrace{\left|\hat{\beta}_i\right|}_{>0} + \underbrace{\left|X_i^T\left(y - X\hat{\beta}\right)\right|}_{=e^\lambda} \\
&\implies q_i = 1.
\end{aligned}
$$

For $i \notin \mathcal{E}$, $\hat{\beta}_i = 0$ since $\operatorname{supp}\hat{\beta} \subset \mathcal{E}$. By (21), we have $\left|X_i^T\left(y - X\hat{\beta}\right)\right| \le e^\lambda$. However, since $i \notin \mathcal{E}$, the inequality is strict

$$
\left|X_i^T\left(y - X\hat{\beta}\right)\right| < e^\lambda
$$

and can be used to solve for $q_i$

$$
\left|\hat{\beta}_i - X_i^T\left(X\hat{\beta} - y\right)\right| = \left|X_i^T\left(y - X\hat{\beta}\right)\right| < e^\lambda \implies q_i = 0.
$$

Finally, for $i \in \mathcal{E} \setminus \operatorname{supp}\hat{\beta}$, $\hat{\beta}_i = 0$ and $\left|X_i^T\left(X\hat{\beta} - y\right)\right| = e^\lambda$ which gives

$$
\left|\hat{\beta}_i - X_i^T\left(X\hat{\beta} - y\right)\right| = \left|X_i^T\left(X\hat{\beta} - y\right)\right| = e^\lambda
$$

and thus $q_i \in [0, 1]$. Putting everything together we get an expression for $q_i$ in terms of $\mathcal{E}$ and $\operatorname{supp}\hat{\beta}$

$$
q_i \in \begin{cases} \{1\} & i \in \operatorname{supp}\hat{\beta} \\ [0, 1] & i \in \mathcal{E} \setminus \operatorname{supp}\hat{\beta} \;, \\ \{0\} & i \notin \mathcal{E} \end{cases} \tag{22}
$$

i.e., $q \in \mathcal{M}$. We proceed to show that $B$ is invertible for all $\lambda \in \mathbb{R}$. Denote $Q := \operatorname{diag}(q)$ for brevity; using the same argument of [58, Theorem 2] involving similarity transformations and continuity, the matrix $B$ is invertible if and only if

$$
\tilde{B} := \operatorname{Id}_p - Q^{1/2}\left(\operatorname{Id}_p - X^TX\right)Q^{1/2} = \operatorname{Id}_p - Q + Q^{1/2}X^TXQ^{1/2}
$$

is invertible. Since $\tilde{B} \succeq \operatorname{Id}_p - Q$, it follows that $\ker\left(\tilde{B}\right) \subset \ker\left(\operatorname{Id}_p - Q\right)$, however $\ker\left(\operatorname{Id}_p - Q\right)$ is a subspace of $W_\mathcal{E} := \operatorname{span}\{e_j : j \in \mathcal{E}\}$ corresponding to $q_j = 1$. Since $q_j = 1 \implies j \in \mathcal{E}$ by (22), the restriction of $\tilde{B}$ to $\ker\left(\operatorname{Id}_p - Q\right)$ is a principal submatrix of (possibly equal to) $X_\mathcal{E}^T X_\mathcal{E}$ which is invertible by assumption. Thus $B$ is invertible and applying Corollary 1 then yields the final result. $\qquad\square$

**Remark 4** Taking $q_i = 1$ for all $i \in \mathcal{E}$ gives a selection of the conservative Jacobian for $\hat{\beta}$ in Proposition 5, for all $j \in \{1, \ldots, p\}$,

$$
[J_{\hat{\beta}}(\lambda)]_j = -e^\lambda\left[\left(X_\mathcal{E}^T X_\mathcal{E}\right)^{-1}\operatorname{sign}\left(X_\mathcal{E}^T\left(y - X\hat{\beta}\right)\right)\right]_j \text{ if } j \in \mathcal{E}, \text{ and } [J_{\hat{\beta}}(\lambda)]_j = 0 \text{ otherwise.}
$$

This corresponds to the directional derivative given by LARS algorithm [28], see also [42]. Alternatively, taking $q_i = 0$ for $i \notin \operatorname{supp}\hat{\beta}$ gives, for all $j \in \{1, \ldots, p\}$,

$$
[J_{\hat{\beta}}(\lambda)]_j = -e^\lambda\left[(X_{\operatorname{supp}\hat{\beta}}^T X_{\operatorname{supp}\hat{\beta}}^{-1})\operatorname{sign}(X_{\operatorname{supp}\hat{\beta}}^T(y - X\hat{\beta}))\right]_j, \text{ if } j \in \operatorname{supp}\hat{\beta}
$$

and $[J_{\hat{\beta}}(\lambda)]_j = 0$ otherwise. This is the weak derivative given by [9]. Both of these expressions are particular selections in $J_{\hat{\beta}}$, which is the underlying conservative field. They agree if $\mathcal{E} = \operatorname{supp}\hat{\beta}$, which holds under qualification assumptions, see for example [10] and references therein.

# D    Results from Section 4

**Theorem 3 (Convergence result)** *Consider minimizing $\ell$ given in* (8) *using algorithm* (10) *under Assumption 1. Assume furthermore the following*

- ***Step size:*** $\sum_{k=1}^{+\infty} \alpha_k = +\infty$ *and* $\alpha_k = o(1/\log(k))$.

- ***Boundedness:*** *there exists* $M > 0$, *and* $K \subset \mathbb{R}^p$ *open and bounded, such that, for all* $s \in (s_{\min}, s_{\max})$ *and* $w_0 \in \mathrm{cl}\, K$, $\|w_k\| \le M$ *almost surely.*

*For almost all $w_0 \in K$ and $s \in (s_{\min}, s_{\max})$, the objective value $\ell(w_k)$ converges and all accumulation points $\bar{w}$ of $w_k$ are Clarke-critical in the sense that $0 \in \partial^c \ell(\bar{w})$.*

**Proof:** We first show that if $w_0$ is taken uniformly at random on $K$ then, almost surely, all iterates $(w_k)_{k \in \mathbb{N}}$ are random variables which are absolutely continuous with respect to the Lebesgue measure. This is essentially a repeating of the arguments developed in [11] for constant step sizes. Assume from now on that $w_0$ is random, uniformly on $K$.

For $i \in \{1, \ldots, N\}$, denoting by $\phi(\cdot, i) \colon \mathbb{R}^p \to \mathbb{R}^p$ the output of backpropagation applied to $\ell_i = g_{i,L} \circ g_{i,L-1} \circ \ldots \circ g_{i,1}$, we have that $x \mapsto \phi(x, i)$ is a selection in the conservative Jacobian (actually conservative gradient) $J_i$. Therefore, using [11, Proposition 1] the sequence $(w_k)_{k \in \mathbb{N}}$ is an SGD sequence in the sense of [11, Definition 2].

Compositions of definable functions and functions implicitly defined based on definable functions are definable. Therefore by Assumption 1, for each $i \in \{1, \ldots, N\}$, $\ell_i$ is locally Lipschitz and definable and thus so is $\ell$. Definable functions are twice differentiable almost everywhere so that [11, Proposition 3] applies. Following the recursion argument in [11, Proposition 2], there exists a set $\Gamma \subset (0, \infty)$ of full Lebesgue measure such that, if $s\alpha_k \in \Gamma$ for all $k \in \mathbb{N}$, each iterate $(w_k)_{k \in \mathbb{N}}$ is a random variable which is absolutely continuous with respect to the Lebesgue measure. We have that

$$\{s \in (s_{\min}, s_{\max}) : \exists k \in \mathbb{N}, s\alpha_k \in (0, \infty) \backslash \Gamma\} = \bigcup_{k=1}^{\infty} \{s \in (s_{\min}, s_{\max}) : s\alpha_k \in (0, \infty) \backslash \Gamma\}$$

is a countable union of null sets and thus a null set, i.e., for almost all $s \in (s_{\min}, s_{\max})$, for all $k \in \mathbb{N}$, $s\alpha_k \in \Gamma$. As a result, for almost all $s$, $w_k$ has a density with respect to the Lebesgue measure for all $k \in \mathbb{N}$.

Conservative gradients are gradients almost everywhere and so there is a full measure set $S$ such that, for all $w \in S$ and all $i \in \{1, \ldots, N\}$, $J_i(w) = \{\nabla \ell_i(w)\}$ [14, Theorem 1]. Combining this with the fact that each element of the sequence is absolutely continuous with respect to the Lebesgue measure, the same argument as in [11, Theorem 1] gives, for almost all $s \in (s_{\min}, s_{\max})$, for every $k \in \mathbb{N}$, almost surely

$$w_{k+1} = w_k - s\alpha_k \nabla \ell_{I_k}(w_k)$$

and

$$\mathbb{E}(w_{k+1}|w_0, \ldots, w_k) = w_k - s\alpha_k \nabla \ell(w_k) = w_k - s\alpha_k \partial^c \ell(w_k).$$

Therefore, the sequence is actually a Clarke stochastic subgradient sequence almost surely (see, for example, [25]) and thus can be analyzed using the method developed in [8]. Indeed, conservativity ensures that $\ell$ is a Lyapunov function for the differential inclusion $\dot{w} \in -\partial^c \ell(w)$, that is decreasing along solutions, strictly outside of $\mathrm{crit}_\ell := \{w \in \mathbb{R}^p, 0 \in \partial^c \ell(w)\}$. Since $\ell$ is definable, the set of its critical values, $\ell(\mathrm{crit}_\ell)$ is finite [13] and thus has empty interior. By [8, Theorem 3.6] and [8, Proposition 3.27], it is then guaranteed that $\ell(\bar{w})$ is constant for all accumulation points $\bar{w}$ of $(w_k)_{k \in \mathbb{N}}$ and that $0 \in \partial^c \ell(\bar{w})$. This occurs almost surely with respect to the randomness induced by $w_0$ and $(I_k)_{k \in \mathbb{N}}$ and therefore it is true with probability one for almost all $w_0$. $\qquad \square$

# E    Results from Section 5

## E.1    Cyclic gradient descent

### E.1.1    Fixed-point formulation

Consider the optimization problem

$$(s_1, s_2) \in \underset{(a,b) \in [0,3] \times [0,5]}{\arg\max} (a+b)(-3x + y + 2). \tag{23}$$

The optimality condition for this problem can be expressed using the fixed-point equation of the projected gradient descent algorithm. Denote for $x, y \in \mathbb{R}^2$, $q_{x,y} : (a,b) \mapsto (a+b)(-3x + y + 2)$; we can verify $(s_1, s_2)$ is solution to (11) if and only if it satisfies the equality

$$\begin{bmatrix} s_1 \\ s_2 \end{bmatrix} = P_{\mathcal{U}} \left( \begin{bmatrix} s_1 \\ s_2 \end{bmatrix} + \nabla q_{x,y}(s_1, s_2) \right) = P_{\mathcal{U}} \left( \begin{bmatrix} s_1 \\ s_2 \end{bmatrix} + \begin{bmatrix} -3x + y + 2 \\ -3x + y + 2 \end{bmatrix} \right).$$

Where $P_{\mathcal{U}}$ is the projection on the set $\mathcal{U} := [0, 3] \times [0, 5]$ which can be implemented as a difference of relu functions

$$P_{\mathcal{U}}(x, y) = \mathrm{relu}(x, y) - \mathrm{relu}(x - 3, y - 5).$$

Let $h : \mathbb{R}^2 \times \mathbb{R} \times \mathbb{R} \to \mathbb{R}^2$ be the function

$$h : (s, x, y) \mapsto P_{\mathcal{U}} \left( \begin{bmatrix} s_1 \\ s_2 \end{bmatrix} + \begin{bmatrix} -3x + y + 2 \\ -3x + y + 2 \end{bmatrix} \right).$$

Then the original problem (23) is equivalent to the fixed point equation $s = h(x, y, s)$. Indeed, we can easily verify the solutions $s : \mathbb{R}^2 \to \mathbb{R}^2$ to (23) are

$$s(x, y) = \begin{cases} \{(0,0)\} & \text{if} \quad -3x + y + 2 < 0 \\ \{(3,5)\} & \text{if} \quad -3x + y + 2 > 0 \\ [0,3] \times [0,5] & \text{if} \quad -3x + y + 2 = 0 \end{cases}$$

which creates a discontinuity for the function $\ell(\cdot, s(\cdot))$, now expressed as

$$\ell(x, y, s(x, y)) = \begin{cases} x^2 + 4y^2 & \text{if} \quad -3x + y + 2 < 0 \\ (x-3)^2 + 4(y-5)^2 & \text{if} \quad -3x + y + 2 > 0 \end{cases}.$$

### E.1.2    Perturbed experiments

Perturbed experiments are done on the following perturbed loss function

$$\ell_\varepsilon(x, y, s) = \left( \frac{1}{4} + \varepsilon_1 \right)(x - s_1)^2 + (1 + \varepsilon_2)(y - s_2)^2$$

$$s \in s_\varepsilon(x, y) := \arg\max \{(a + b)(-(3 + \varepsilon_3)x + y + 2 + \varepsilon_4) : a \in [0, 3 - \varepsilon_5], b \in [0, 5 - \varepsilon_6]\}$$

with $\varepsilon_1, \ldots, \varepsilon_6$ the perturbations. In Figure 2b, we consider several realizations of independent Gaussian variables $\varepsilon_1, \ldots, \varepsilon_6 \sim \mathcal{N}(0, \sigma^2)$ with $\sigma^2 = 0.05$; despite this added noise, the unwanted dynamics persist.

### E.1.3    Conic canonicalization

Let $c \in \mathbb{R}^2$ be a parameter vector and consider the problem

$$\max_{x \in [0,3] \times [0,5]} c^T x.$$

It can be formulated as a cone program (P) and its dual (D):

$$\text{(P)} \quad \inf \quad c^T x$$
$$\text{subject to} \quad Ax + s = b$$
$$s \in \mathcal{K}$$

$$\text{(D)} \quad \inf \quad b^T y$$
$$\text{subject to} \quad A^T y + c = 0 \qquad (24)$$
$$y \in \mathcal{K}^*,$$

where

$$A = \begin{bmatrix} \mathrm{Id}_2 \\ -\mathrm{Id}_2 \end{bmatrix} \text{ and } b = \begin{bmatrix} 3 \\ 5 \\ 0 \\ 0 \end{bmatrix}.$$

Let $(x, y, s)$ be a solution to the cone program (24) where $x$ is the primal variable, $y$ is the dual variable, and $s$ the primal slack variable. Then it follows from (6) that a solution $z$ to $\mathcal{N}(z, c) = 0$ is obtained by $z = (x, y - s)$. For $c = (0, 0)$, the solutions are $x \in [0, 3] \times [0, 5]$, $s = b - Ax$, and $y = (0, 0, 0, 0)$, hence the uniqueness assumption for Proposition 4 is not satisfied.

### E.1.4 A chaotic dynamics in $\mathbb{R}^4$

We combine two cycles of the previous example into a gradient dynamics in $\mathbb{R}^4$. To perform this, we consider a block-separable sum of the same function where we add a scaling parameter $\eta > 0$:

$$g : (x, y, z, w) \mapsto f(x, y) + \eta f(z, w).$$

This will combine the two cycles but the parameter $\eta$ will make one cycle "faster" than the other. Projecting the path of the gradient descent on the variables $(y, z)$ we obtain a chaotic dynamics filling the space as the number of iterations increases.

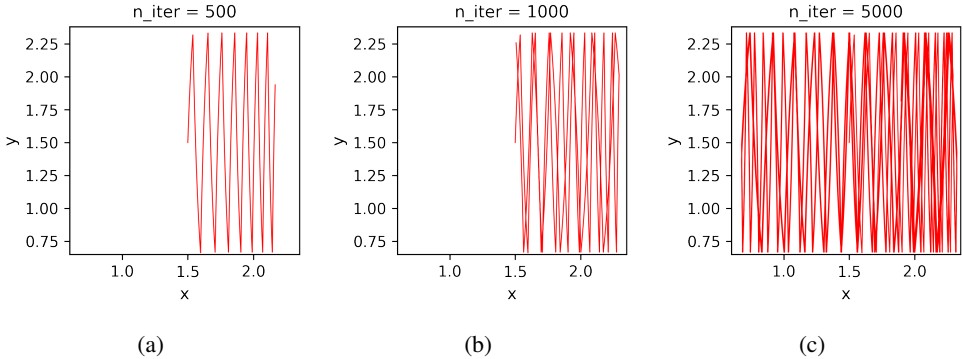

(a)       (b)       (c)

Figure 5: Gradient path after (a) 500, (b) 1000 and (c) 5000 iterations.

## E.2 Lorenz-like attractor

### E.2.1 Objective function is a quadratic form

Set $u = (x, y, z)$, then

$$u^T F(u) = \sigma x(y - x) + xy(\rho - z) - y^2 + xyz - \beta z^2$$
$$= -\sigma x^2 - y^2 - \beta z^2 + (\sigma + \rho)xy$$
$$= \frac{1}{2} u^T H u$$

where $H = \begin{bmatrix} -2\sigma & \sigma + \rho & 0 \\ \sigma + \rho & -2 & 0 \\ 0 & 0 & -2\beta \end{bmatrix}$.

For $(\sigma, \rho, \beta) = (10, 28, \frac{8}{3})$, $g$ has for unique critical point $(0, 0, 0)$ which is a strict saddle-point.

## E.3 License of assets used

All assets used: cvxpy, cvxpylayers, and JAX were released under the Apache License, Version 2.0, January 2004, `http://www.apache.org/licenses/`.