# OpenReview forum: "Nonsmooth Implicit Differentiation for Machine-Learning and Optimization"
_NeurIPS.cc/2021/Conference — NeurIPS 2021 Poster_

### Official Review · Reviewer_eaLv · 2021-07-15

**Rating:** 6
**Confidence:** 3

**Summary:**

This work provides theoretical foundations for nonsmooth implicit differentiation based on conservative Jacobians. Specifically, it provides conditions under which implicit differentiation is compatible with backpropagation and first-order optimization algorithms. The authors then study the applications of the implicit differentiation algorithm, including deep equilibrium problems, conic optimization layers, and hyperparameter optimization problems for Lasso. The specific conditions and the forms of the Jacobians are also provided. After that, the authors obtain the convergence guarantees for first-order optimization algorithms. Finally, two examples -- a cyclic gradient dynamic via optimization layer and a Lorenz-like attractor via implicit differentiation – are used to demonstrated that the optimization may fail when the conditions are not satisfied.

**Limitations And Societal Impact:**

This work is purely theoretical and does not involve any applications that may lead to negative social impact.

**Main Review:**

The paper is well-written and is written in a very rigorous way. The related concepts are well-defined and the failure cases are clearly demonstrated.

The precise conditions for applicability of backpropagation and first-order optimization can be useful for many machine learning applications, which is a solid contribution to the community.

As a small suggestion for improvement: If I was writing this paper, I would put the current section 5, i.e., numerical experiments, in the introduction or section 2, as part of the motivation for this work. The current section 2, i.e., the main results, is rigorous but abstract. Describing it with some examples can make it more concrete for the readers. Besides, the current numerical experiments are more like two examples than real ML experiments.


**Time Spent Reviewing:**

3

---

> ### Author Response · Authors · 2021-08-05
> **Response to reviewer eaLv:**
>
> Many thanks for your report.
>
> ### Numerical experiments are more like two examples than real ML experiments:
>
> As explained in answer to 2ovw, we fully agree: they are counterexamples, and they have been designed with maximal simplicity (simple structure, low dimension). The goal was to show failures of the gradient method when our main assumption of "invertibility" is absent.
>
> The first example shows an absurd behavior in dimension 2. In dimension 3, we present a failure having a chaotic-like behavior. The fact that this holds at a very small scale makes our point much stronger.
>
>
> ### Put Section 5 to the forefront
>
> Thanks for the suggestion. This was actually the purpose of having Figure 3 (b) at the beginning of the paper. However, as the main contribution is theoretical and the examples are only illustrative (not real-life ML models), we decided not to put all the details to the forefront. Still, we are ready to consider a change.

---

> > ### Comment · Reviewer_eaLv · 2021-09-01
> > **Thanks for the rebuttal**
> >
> > Thanks for the response. I have no major issues accepting this paper. I will keep my score as it is then.

---

### Official Review · Reviewer_ZCf8 · 2021-07-16

**Rating:** 6
**Confidence:** 1

**Summary:**

The paper introduces a nonsmooth implicit differentiation theorem via conservative Jacobians, which is compatible with algorithmic differentiation. The authors then apply this theorem to optimize over implicit differentiation problems with gradient descent and proves a convergence result of the algorithm.

**Limitations And Societal Impact:**

I do not see any limitations or potential negative societal impact of the work.

**Main Review:**

**Quality**: I did not check the theorem statements/proofs, though the logical progression of the theorems makes sense to me.

**Clarity**: The paper is very clear and well-motivated. I especially like the way that the authors use explicit (counter)examples to motivate their discussions throughout the text.

**Originality and Significance**: Even though I have not worked on nonsmooth implicit problems in the past, I do expect that such problems can have very different behavior from smooth ones, and the framework proposed in the paper is a good step going forward in this direction. From a practical perspective, I would say that the experiments performed in the paper are rather of small scale, and I would be curious if the authors can implement any one of the three deep learning examples presented in the paper to see how the time complexity of such methods grow with large-scale problems. Since my field of research is only marginally related to this, I do not have much to comment on the originality/impact of the paper.

**Time Spent Reviewing:**

6

---

> ### Author Response · Authors · 2021-08-05
> **Response to reviewer ZCf8:**
>
> Many thanks for your report.
>
> ### The experiments are small scale
>
> This comment concurs with a remark of reviewer 2ovw. The proposed experiments only illustrate potential pathologies related to lack of invertibility in implicit differentiation. These are thus negative results, so the fact that they hold at a very small scale makes the statement stronger. Whence the results are « examples », not real-world machine learning experiments. Investigating how often these pathologies actually occur in practice is very interesting but beyond the scope and constitutes a topic of future work.
>
> We will explain this more clearly in the numerical section.
>
> ### Implementation of the three examples:
>
> Our first motivation is to provide a theory for a practice that is used at scale in the cited literature. The proposed models are already implemented with dedicated numerical libraries with the possibility to treat reasonable scale problems (cf, DEQ libraries, cvxpylayers, etc ...).

---

### Official Review · Reviewer_gMrk · 2021-07-16

**Rating:** 7
**Confidence:** 3

**Summary:**

Conservative Jacobians provide a formalism that justifies the application of standard AD techniques even when the computational graph contains nonsmooth Lipschitz elemental functions such as relu. An element of the conservative Jacobian can be computed by picking any element of the Clarke Jacobian for the elemental functions and applying the chain rule. In that sense the formalism of conservative Jacobians has a significant practical benefit compared to previously proposed approaches for nonsmooth AD, such as e.g. [1] and [2], because no directional information needs to be propagated (there is no qualification).

In this paper the authors present the natural application of the conservative Jacobian formalism to implicit differentiation and highlight the necessity of invertibility as analogous to the classical implicit function theorem.

The authors highlight some applications of implicit differentiation in the area of machine learning:
 - Monotone implicit layers
 - Embedded optimization problems such as optimization layers and hyperparam optimization for L1 regularized optimization problems

If the optimality condition for an embedded optimization problem can be formulated as a nonsmooth implicit function with the assumed Lipschitz composed structure then the authors' implicit function theorem applies.

In some cases it can be possible to prove that the invertibility condition is guaranteed.

The authors give a theorem showing that gradient descent with a sufficient step size decay converges to Clarke stationary points when conservative gradients are used. This theorem is not related to the implicit function theorem and also holds for non-implicit conservative differentiation.

The authors show that conservative gradient dynamics for nonsmooth functions can be cycling and even chaotic in the case the invertibility assumption is not satisfied and demonstrate the resulting discrete gradient dynamics in numerical experiments to highlight the relevance of the invertibility assumption of their implicit function theorem in practical examples.

[1] Branch-locking AD techniques for nonsmooth composite functions and nonsmooth implicit functions
[2] Provably Correct Automatic Subdifferentiationfor Qualified Programs

**Limitations And Societal Impact:**

Yes

**Main Review:**

Strong points:

- The paper gives theoretic justification to methods that are already used in practice.
- The paper provides examples for use cases that show that the theory is somewhat relevant to practical applications (in machine learning)
- The authors formally extend on a theory that has been picked up by different authors for analyzing nonsmooth functions (in machine learning applications)
- The theory and derivations are clean and of high quality and seems relevant

Weak points:

- The paper itself does not provide a good motivation and intuition of how the conservative Jacobian works. As a reader that is not familiar with the conservative Jacobian formalism it is probably necessary to go to the cited literature to gain clarity.
- The paper does not cite other approaches for nonsmooth differentiation in the context of AD, explain how they apply in the context of implicit differentiation, and what the benefits of the conservative Jacobian formalism are over these other approaches (see e.g. [1], [2] cited in the summary).
- From the toy numerical examples provided to show that invertibility can be an issue it is not immediately clear whether these problems occur in any practical machine learning applications.

Originality: Although the application of the conservative Jacobian formalism to implicit differentiation follows naturally the associated analysis of the implications and applications to e.g. bilevel optimization is original work.

Quality: The presented theory is high quality. The numerical experiments are only toy examples.

Clarity: Clarity can be improved in the motivation and introduction of the formalism. The presentation of the implicit function theorem result itself, example use cases and convergence theory for gradient descent are clear.

Significance: The paper is probably not immediately significant to machine learning practitioners but the ideas in the paper advance the state of formal understanding of practices that work in machine learning in practice and is, therefore, significant.

I recommend to accept the paper because the authors present and analyze the consequences of an implicit function theorem that follows naturally from a theory for nonsmooth AD that seems highly relevant to what is used in the practical machine learning world.

Recommendations:
- Improve the initial explanation of the conservative differentiation formalism as it applies to AD as used in ML frameworks
- Contrast your results to other nonsmooth AD approach to highlight the pros / cons

An example of implicit differentiation via elements of the conservative derivative for an implicit function that does not satisfy the invertibility condition is the following:

Let

0 = x - (relu(y) - relu(-y)) = g(x, y(x))

where since relu(y) - relu(-y) = y we have y(x) = x and hence y'(x) = 1 everywhere.

But since at x = 0, y = 0 a valid element of the Clarke Jacobian of relu(y) is relu'(y) = 0 by AD and implicit function theorem at x = 0 we can have

y'(x) = -dg/dy^{-1} * dg/dx = -0^{-1} * 1

i.e. in such a case the invertibility condition is not satisfied.

With a "limiting Jacobian" approach as e.g. suggested by [2] we can associate the direction dir_y = 1 and which would lock the derivatives to

relu'(y; dir_y) = 1
relu'(-y; dir_y) = 0

or alternatively for dir_y = -1 we have

relu'(y; dir_y) = 0
relu'(-y; dir_y) = -1

and so in both cases we have

dg/dy(.; dir_y) = (relu'(y; dir_y) - relu(-y; dir_y)) = 1

which avoids the non-invertibility.

- I'm not sure whether the convergence result for gradient descent should be in the paper as it is already dense and the result is not specific to the implicit function theorem.

Nits:
- Right of Figure 1 looks very similar to Figure 3 (b), is it necessary to have both?
- For the first example you have
x = tanh(z) + relu(-z) + z - relu(z)
= tanh(z) + z + max(0, -z) - max(0, z)
= tanh(z) + z - z
= tanh(z)
which seems to imply that z = tanh^{-1}(x) instead of z = tanh(x) like you write.
The legend of figure one (invtanh) suggests that's what you had in mind anyways.


**Time Spent Reviewing:**

5

---

> ### Author Response · Authors · 2021-08-05
> **Response to reviewer gMrk:**
>
> Many thanks for your report.
>
> ### How conservative Jacobian work
>
> The main message is that backprop does not compute Clarke Jacobians in general. However, backprop can be seen as an oracle for a conservative Jacobian. This conservative Jacobian can be obtained by considering all possible outputs of backpropagation. We will expand on this in Section 2.
>
>
> ### Bibliography on nonsmooth algorithmic differentiation (AD)
>
> As mentioned in response to reviewer 2ovw, we will add a section (possibly in the appendix) discussing alternative approaches. In addition, the proposed references will be added to this discussion.
>
> * Provably Correct Automatic Subdifferentiation for Qualified Programs: this work highlights problems of nonsmooth AD. This is solved by requiring a qualification assumption, delicate to check in practice. Under this assumption, the authors propose an AD algorithm to compute an element of the Clarke Jacobian. Conservative Jacobians constitute an opposite view on nonsmooth AD: the notion of a derivative is weaker, but it does not require further assumption (qualification). As for the implementation, it is the one currently made by popular libraries.
> Note also that it is not clear how the proposed qualification condition would apply to implicit differentiation.
>
> * Branch-locking AD techniques for nonsmooth composite functions and nonsmooth implicit functions.
> This interesting work is limited to lexicographic derivatives (which are directional derivatives). A similar reference on directional derivatives was given by reviewer 2ovw (Robinson's paper). Currently, directional derivatives are of less interest for optimization in DL context as most algorithms require vectors descent directions. Furthermore, algorithmic differentiation based on directional derivatives is not implemented in most numerical libraries.
>
>
> ### Do pathologies occur in practice?
>
> This is indeed an interesting question. But, unfortunately, we do not understand this issue at this stage, and it is certainly a matter for future research.
>
> ### Recommendation on conservative differentiation
>
> We will indeed expand on the fact that backprop is an oracle for conservative Jacobians; this is the key aspect from a practical point of view.
>
> ### Recommendation regarding alternative nonsmooth AD approaches
>
> We will also add more discussion about the bibliography in an additional section in the Appendix.
>
> ### Example given by the reviewer about invertibility:
>
> This example is very interesting; we warmly thank the referee for this. Interactions with our work should indeed be investigated. However, it is based on directional derivative calculus, as in the paper of Robinson (mentioned by reviewer 2ovw) and Khan's paper. This does not fit our objectives of a calculus compatible with ``TensorFlow's AD », i.e., with a straightforward compositional calculus (optimization, training with existing numerical libraries).
>
> ### Further comments
>
> We decide to keep the convergence result to illustrate the relevance of conservative gradients, which is a non-standard concept for the ML community.

---

### Official Review · Reviewer_2ovw · 2021-07-16

**Rating:** 6
**Confidence:** 3

**Summary:**

This paper surveys and summarizes the theoretical support for performing implicit differentiation in the broader, not-necessarily smooth settings. The derived formulas and theories, importantly, are compatible with AD. The authors have established the generalization of their conservative-Jacobian-based approach to multiple scenarios, including equilibrium networks, optimization layer, etc.

**Limitations And Societal Impact:**

As this is a theory paper, the authors have discussed the limitations by clearly stating the assumptions.

**Main Review:**

Overall, I believe this paper rigorously explored a gap that prior works on implicit deep learning have missed, which is the non-smooth structures. To that end, I think this paper is appropriate for the NeurIPS community, and offers a solid & sound theoretical support for the recent progress in implicit networks (which is obviously needed, and is why I recommend acceptance). However, I'm also simultaneously concerned about 1) the novelty of the conclusion; and 2) the practicality of theoretical results.

Pros:
- Rigorous math and well-stated assumptions, conditions and results.
- Tackles a missing gap by the prior works in this direction (i.e., IFT applied in non-smooth settings)
- The theory captures a relatively wide variety of scenarios, including equilibrium networks, hyperparameter optimization (bi-level optimization), and optimization layer.

Cons:
- The technique used for the non-smooth analysis is not new (conservative Jacobians), and there have been (as the authors acknowledged) plenty of prior explorations of non-smooth implicit differentiation in the literature (e.g., "An implicit function theorem for nonsmooth function" by Robinson; "Implicit functions and sensitivity of stationary points" by Jongen et al. (the authors didn't cite this?); the monotone DEQ paper also briefly discussed Clarke generalized Jacobian).
- The results derived completed the theoretical picture of various implicit ML methods but has limited practical implications (unless the function is extremely ill-posed). For instance, the result for monotone DEQ (proposition 3) is not different from theorem 2 of [Winston & Kolter 2020] (but better characterizes the condition and assumption for a more general applicability). As another example, the hyperparameter optimization discussion has been centered around the LASSO type problem.
- The experiment has a limited scope and is on low dimension.

I'm also particularly curious about the case where an equilibrium network does not have a unique fixed point. In particular, most likely we can guarantee uniqueness locally (e.g., by making the layer locally contractive, and then making sure we initialize the fixed point estimate well) but not globally in the entire R^n space. How will the theory change in this case (and broadly, when the layer is not monotone)?

---------
---------

### Post-rebuttal

I've read the authors' responses and would like to thank the authors for the clarification. I don't have any major issue with accepting this paper, but at the same time (as the authors acknowledge) there is still limited practicality to the analysis presented here. I urge the authors to include the references I pointed out in the review.

**Time Spent Reviewing:**

7

---

> ### Author Response · Authors · 2021-08-05
> **Response to reviewer 2ovw**
>
> Many thanks for the report.
>
> ### Bibliography on nonsmooth implicit differentiation:
>
> Many thanks for the useful references; we will, of course, cite them and detail the difference with our work (in an additional section in the Appendix).
>
> * Robinson's contribution is twofold. First, his Theorem 3.2 ensures the existence of implicit functions without calculus, generalizing a theorem of Clarke. Secondly, Corollary 3.4 provides calculus rules but is limited to directional derivatives requiring strong assumptions (for which we are not aware of any easily verifiable condition). More decisively, this calculus is not compatible with AD as implemented on standard libraries.
> Our paper proposes a matrix-based calculus that fits the implementation of major deep learning libraries and works for all semi-algebraic problems without any other assumptions than the usual invertibility.
>
> * The paper of Jongen et al. generalizes the implicit function theorem of Clarke for a function G approximating a function F (in a natural norm) for which the conditions of the implicit function theorem hold. Unfortunately, there is no calculus which is our objective here.
>
> * The monotone DEQ paper indeed mentions Clarke Jacobian, but the inversion formula is applied without justification. It is an issue since we show that the inverse is not a Clarke Jacobian in general. So the calculus they used is based on objects with unknown variational meanings. Consequently, the mathematical part of their result does not provide full guarantees on the training of nonsmooth problems with AD.
> Contrarywise our framework shows that inverting Clarke Jacobian is meaningful when working with conservative Jacobians in a semialgebraic/definable context. More importantly, we establish that our implicit function theorems agree with the current implementation of AD, which allows obtaining general convergence guarantees for the training of NN.
>
> ### Limited practical implications
>
> * As already mentioned,  nonsmooth AD through matrix inversion is impossible if one restricts oneself to the Clarke Jacobian framework.
> We overcome this problem and provide an adequate framework for justifying practical applications, as training DEQs, hyperparameter optimization, or using conic programming layers.
>
> * For hyperparameter tuning, we focused on Lasso for illustration purposes; actually, we are currently studying general convex regularized problems, it is beyond the scope of this paper.
>
> * For the conic programming applications, note that the only available results were when the residual map is differentiable (which is impossible to enforce in general). Therefore, we generalize this aspect considerably.
>
> ### The experiments have limited scope
>
> Our results are indeed "examples" (even counterexamples to the good behavior of the "nonsmooth gradient method") and not real machine learning experiments.
>
> The precise goal was to show the failure of the gradient method when our main assumption of "invertibility" is absent. Therefore our examples are chosen as simple as possible and with the lowest possible dimensions.
>
> The first example shows failure in dimension 2. Then, we go to dimension 3 to show that failure may be close to ... a chaotic behavior (which only occurs in dimension greater than 3)! The fact that this holds at a very small scale makes the statement stronger, not weaker.
>
> Investigating how often these pathologies actually occur in practice is beyond the scope of this paper.
>
> ### No unique fixed point in equilibrium networks
>
> The example in Figure 2 can actually be implemented as training a simple equilibrium network involving the relu activation function (Appendix E.1.1). This is done by observing that optimality is equivalent to being a fixed point of the projected gradient mapping.
> The theory does not extend directly to multiple fixed points because the objective function may become discontinuous, generating unexpected gradient dynamics (our first example shows a cycle far from the critical region).
> Whence, without uniqueness and new assumptions, one may encounter pathological training dynamics. We will emphasize this in the numerical section.

---

> ### Author Response · Authors · 2021-08-25
> **Reponse to the Post-Rebuttal**
>
> Thanks again for the report. As mentioned in our response, we will be sure to add the suggested references and expose their connection to our work.

---

### Decision · Program_Chairs · 2021-09-27

**Decision:**

Accept (Poster)

**Comment:**

This paper provided the justification of the implicit differentiation in the non-smooth settings with conservative Jacobians. The proposed method is also compatible with the standard AD. The authors also established the connection of the conservative Jacobians with the Clarke Jacobian. The authors finally apply the results to multiple machine learning problems, e.g., deep equilibrium networks, optimization layers, and bi-level optimization. The paper is well-written and easy-to-follow. I believe this paper will be of interest to the wide range of NeurIPS community.

There are still several minor issues need to be address, especially the discussion to the related work in literature (Reviewer gMrk, 2ovw) and more comprehensive empirical experiments (Reviewer eaLv).